# Calcium-mediated rapid movements defend against herbivorous insects in *Mimosa pudica*

Takuma Hagihara [1,7], Hiroaki Mano [2,3,4,7], Tomohiro Miura[1],
Mitsuyasu Hasebe [2,3] ✉ & Masatsugu Toyota [1,5,6] ✉

Animals possess specialized systems, e.g., neuromuscular systems, to sense the environment and then move their bodies quickly in response. *Mimosa pudica*, the sensitive plant, moves its leaves within seconds in response to external stimuli; e.g., touch or wounding. However, neither the plant-wide signaling network that triggers these rapid movements nor the physiological roles of the movements themselves have been determined. Here by simultaneous recording of cytosolic $Ca^{2+}$ and electrical signals, we show that rapid changes in $Ca^{2+}$ coupled with action and variation potentials trigger rapid movements in wounded *M. pudica*. Furthermore, pharmacological manipulation of cytosolic $Ca^{2+}$ dynamics and CRISPR-Cas9 genome editing technology revealed that an immotile *M. pudica* is more vulnerable to attacks by herbivorous insects. Our findings provide evidence that rapid movements based on propagating $Ca^{2+}$ and electrical signals protect this plant from insect attacks.

Plants use electrical signals that travel long distances to share local stimulus information with distant sites[1]. *Mimosa pudica*, commonly called touch-me-not, shame or sensitive plant, perceives a variety of stimuli and almost immediately moves its leaves, yet it lacks the neurons and muscles that would underlie such movements in animals[2–4]. Non-wounding stimuli, e.g., mechanical touch, cold shock, or electrical stimulation, applied to the *M. pudica* leaf, generate a rapid depolarization of the membrane potential, i.e., an action potential (AP), propagating toward the motor organ or pulvinus at the bases of leaflets, rachillae, and petioles[5,6]. Wounding stimuli, e.g., cutting or burning, generate both APs and a subsequent long-lasting delayed depolarization, named a variation potential (VP)[5,7,8]. When the electrical signals arrive, the pulvinar cells of the contractile (extensor) side shrink due to water efflux (loss of turgor pressure), instantaneously folding up the leaflets and dropping the petiole downwards[9,10]. Although numerous studies have assumed physiological roles of these rapid movements, e.g., being unnoticed against the dark ground[2], startling insects[11],

exposing thorns[12], and giving the appearance of a less voluminous meal[13], clear evidence supporting these theories thus far does not exist. Furthermore, many gaps remain in our knowledge of the mechanisms underlying this rapid movement and the plant-wide signal network used to trigger it.

Here by simultaneous measuring of cytosolic $Ca^{2+}$ and electrical signals, we show that rapid leaf movements mediated by changes in $Ca^{2+}$ coupled with action and variation potentials protect *M. pudica* from herbivorous insect attacks.

## Results

### $Ca^{2+}$ signals trigger rapid movements

Both the electrical signals and leaf movement are known to attenuate when the extracellular $Ca^{2+}$ concentration is reduced in *M. pudica*[14,15], implying a role for $Ca^{2+}$ signaling in this process. To determine whether rapid movements might be regulated by $Ca^{2+}$ and electrical signals, we created transgenic *M. pudica* expressing the genetically encoded $Ca^{2+}$

[1]Department of Biochemistry and Molecular Biology, Saitama University, Saitama 338-8570, Japan. [2]Division of Evolutionary Biology, National Institute for Basic Biology, Okazaki 444-8585, Japan. [3]School of Life Science, Graduate University for Advanced Studies (SOKENDAI), Okazaki 444-8585, Japan. [4]JST, PRESTO, Saitama 332-0012, Japan. [5]Suntory Rising Stars Encouragement Program in Life Sciences (SunRiSE), Suntory Foundation for Life Sciences, Kyoto 619-0284, Japan. [6]Department of Botany, University of Wisconsin, Madison, WI 53706, USA. [7]These authors contributed equally: Takuma Hagihara, Hiroaki Mano. ✉e-mail: mhasebe@nibb.ac.jp; mtoyota@mail.saitama-u.ac.jp

indicator GCaMP6f[16] and visualized the spatiotemporal dynamics of the cytosolic $Ca^{2+}$ concentration ($[Ca^{2+}]_{cyt}$) in real time. GCaMP6f showed that mechanical touch with forceps causes a rapid but spatially localized $[Ca^{2+}]_{cyt}$ increase in various organs, such as floral buds and roots (Supplementary Fig. 1), without affecting the physiological response of leaf movement (Supplementary Fig. 2). In contrast, touch and wounding a leaflet with scissors induced $[Ca^{2+}]_{cyt}$ increases at the base of leaflets (tertiary pulvinus) in parallel with leaflet movement, which was sequentially triggered along a rachilla (Fig. 1a, b; Supplementary Movies 1 and 2). High-speed $Ca^{2+}$ imaging revealed that wound-induced $[Ca^{2+}]_{cyt}$ increase preceded leaflet movement by up to 0.15 s (Fig. 1c, e and Supplementary Movie 3). Pretreating *M. pudica* leaves with a $Ca^{2+}$ channel blocker, $La^{3+}$, prevented both the $[Ca^{2+}]_{cyt}$ increase and movement in response to wounding (Fig. 1d, f and Supplementary Movie 4). Therefore, a $[Ca^{2+}]_{cyt}$ increase in the pulvinus is correlated with rapid leaf movement.

We also analyzed the $[Ca^{2+}]_{cyt}$ propagation pathway from the wound site to pulvini on a rachilla by monitoring GCaMP6f fluorescence from the abaxial (lower) side of the leaves. Wounding by scissors immediately elicited a $[Ca^{2+}]_{cyt}$ increase at the wound site, which subsequently propagated in a leaflet vein at $1.31 \pm 0.17$ mm/s ($n = 6$; Fig. 2a–c and Supplementary Movie 5). The traveling $[Ca^{2+}]_{cyt}$ increase was followed by an abrupt rise in $[Ca^{2+}]_{cyt}$ at the tertiary pulvinus and leaflet movement (Fig. 2a–c and Supplementary Movie 5), further propagating in a rachilla bidirectionally (Fig. 2a, d and Supplementary Movie 5). The $[Ca^{2+}]_{cyt}$ signature on the rachilla was bimodal (Fig. 2d), with the first $[Ca^{2+}]_{cyt}$ peak propagating basipetally at $3.11 \pm 0.41$ and acropetally at $2.65 \pm 0.38$ mm/s ($n = 10$ each). Pretreatment of $La^{3+}$ and a $Ca^{2+}$ chelator, EGTA, retarded propagation of the $[Ca^{2+}]_{cyt}$ increase as well as the pulvinar $[Ca^{2+}]_{cyt}$ increase, whereas a slight $[Ca^{2+}]_{cyt}$ change was observed at the wound site (Supplementary Fig. 3 and Supplementary Movies 6 and 7). These results suggest that $[Ca^{2+}]_{cyt}$ acts as a long-distance rapid signal triggering leaf movements in wounded *M. pudica*.

## $Ca^{2+}$ and electrical signals are spatiotemporally coupled

Since non-wounding stimuli trigger only an AP[7] and mechanical wounding generates both an AP and VP propagating in a rachilla toward the pulvinus[5], we investigated the spatiotemporal relationship between $[Ca^{2+}]_{cyt}$ transmission and the electrical signals. Touching a pinna tip evoked single-peak $[Ca^{2+}]_{cyt}$ signal and AP with leaflet movements in a rachilla (Fig. 2e, g, i). Wounding a leaflet triggered propagation of bimodal $[Ca^{2+}]_{cyt}$ and electrical signals consisting of an AP and VP (the first and second peaks, respectively) in a rachilla (Fig. 2f, h, j; Supplementary Fig. 4 and Supplementary Movie 8). The touch-induced $[Ca^{2+}]_{cyt}$ signal and AP propagated on the rachilla at $5.87 \pm 0.75$ and $5.52 \pm 0.43$ mm/s, respectively (Supplementary Table 1). The wound-induced $[Ca^{2+}]_{cyt}$ signal and AP transmitted on the rachilla at $4.13 \pm 0.45$ and $4.27 \pm 0.41$ mm/s, respectively (Supplementary Table 1).

Moreover, in contrast to the control experiments (Supplementary Figs. 5a, b, e, f and 6a, b, e, f, Supplementary Movies 9, 10, 13, 14), $La^{3+}$ and EGTA pretreatments inhibited the bidirectional propagation of the $[Ca^{2+}]_{cyt}$ increases and APs/VPs (Supplementary Figs. 5c, d, g, h and 6c, d, g, h, Supplementary Movies 11, 12, 15, 16), suggesting that the long-distance transmission of $Ca^{2+}$ changes is spatiotemporally coupled with the AP and VP, triggering rapid leaf movements in touched or wounded *M. pudica*.

## Rapid movements defend plants against insects

The physiological significance of the rapid movements of *M. pudica* is a long-standing mystery in plant science but have been speculated to relate to an antiherbivory defense response[2,11–13]. To ask if these rapid movements do indeed serve as a defense response to wounding by insect herbivory, we used $La^{3+}$-treated leaves that did not respond to wounding or touch (Fig. 3a, b; Supplementary Figs. 7 and 8a, b).

Grasshopper herbivores stayed and fed on the $La^{3+}$-treated leaves more than on the control leaves (Fig. 3d). $La^{3+}$-treated leaves lost 38.0% in weight after this feeding assay, which was ~2-fold higher consumption than that of the control leaves [18.9% ($n = 14$ each); Fig. 3f]. Consistent with this result, the total residence time of grasshoppers was $75.3 \pm 11.3$ min on the control leaves and $161.2 \pm 23.1$ min on the $La^{3+}$-treated leaves ($n = 14$ each; Fig. 3h). To exclude the possibility that $La^{3+}$ might change the palatability of leaves and influence the appetite of grasshoppers, we produced an immotile *M. pudica* in a different way, using CRISPR-Cas9 genome editing technology. *ELONGATED PETIOLULE1* (*ELP1*), a putative transcription factor containing the ASYMMETRIC LEAVES2 (AS2)/LATERAL ORGAN BOUNDARIES (LOB) domain, is necessary for pulvinus development in *Medicago truncatula*[17]. Knocking out the homologous genes, *ELP1B1/ELP1B2*, rendered *M. pudica* immotile due to the lack of pulvini (Supplementary Figs. 9 and 10). Although *elp1b1elp1b2* leaves propagated electrical signals upon wounding (Supplementary Fig. 11), the leaves did not move (Fig. 3c and Supplementary Figs. 7, 8c, d, 12a), indicating that the wound signaling pathway is unaffected by the loss of gene function and motility. As seen in the immobile $La^{3+}$-treated leaves, the immotile *elp1b1elp1b2* leaves lost 30.7% in weight by grasshopper attacks, which was approximately double that of wild-type (WT) leaves [15.3% ($n = 12$ each); Fig. 3e, g], and the total residence time of grasshoppers was $80.2 \pm 12.4$ min on the WT leaves and $142.7 \pm 22.4$ min on *elp1b1elp1b2* leaves ($n = 12$ each; Fig. 3i). This effect was not limited to grasshoppers since we also used a generalist caterpillar and obtained similar results (Supplementary Fig. 12b–e). These observations indicate that rapid leaf movements defend against attack by a range of herbivorous insects reducing tissues lost and limiting the time the insects spend on the *M. pudica* leaves.

## Feeding triggers $Ca^{2+}$-mediated movement

To test if insect feeding induced $Ca^{2+}$-based movement, we monitored $[Ca^{2+}]_{cyt}$ changes during grasshopper attack. A grasshopper chewing on leaflets caused a $[Ca^{2+}]_{cyt}$ increase in the tertiary pulvini and rapid leaflet movements (Fig. 4 and Supplementary Movies 17 and 18). After the second feeding (Fig. 4, 60 s), the grasshopper's leg was pinched by folded leaflets, and the grasshopper repetitively shook its leg and moved away from this leaf (Supplementary Movie 18).

## Discussion

Plants activate local and systemic defense responses within minutes to hours of insect contact, wounding, or herbivory, e.g., through production of the phytohormones ethylene and jasmonate, priming non-damaged regions to mount pre-emptive defenses[18–21]. Here, we demonstrated that *M. pudica* has a $Ca^{2+}$/electrical signal-induced rapid defense response that actively repels predators. Compared to the hormone-based defenses, the motion-based defense is much faster, being activated and propagated throughout the plant body within seconds. This speed likely helps protect the plants from immediate insect attack. Touch triggered a single-peak $[Ca^{2+}]_{cyt}$ change and AP in the rachilla, sequentially inducing $[Ca^{2+}]_{cyt}$ increases at pulvini and subsequent leaflet movements (Figs. 1a, c, e, and 2e, i). Thus, *M. pudica* might sense herbivore contacts to activate motion-based defenses before leaves are damaged, but leaflet movements are restricted within the touched pinna because the AP cannot propagate over the secondary pulvinus toward distant pinnae[7]. Wounding elicits a VP that can pass through primary and secondary pulvini[5,7], triggering defenses in both the local and systemic leaves. However, this model should be validated in the field.

Our model is based on pharmacological methods with possible side effects; e.g., $La^{3+}$ amplifies a secondary $Ca^{2+}$ increase induced by 0 mM $K^+$ medium in *Arabidopsis thaliana* roots[22], causes plasma membrane depolarization that is reduced by raising the extracellular $Ca^{2+}$ concentration in *Neurospora crassa*[23], and induces phosphate

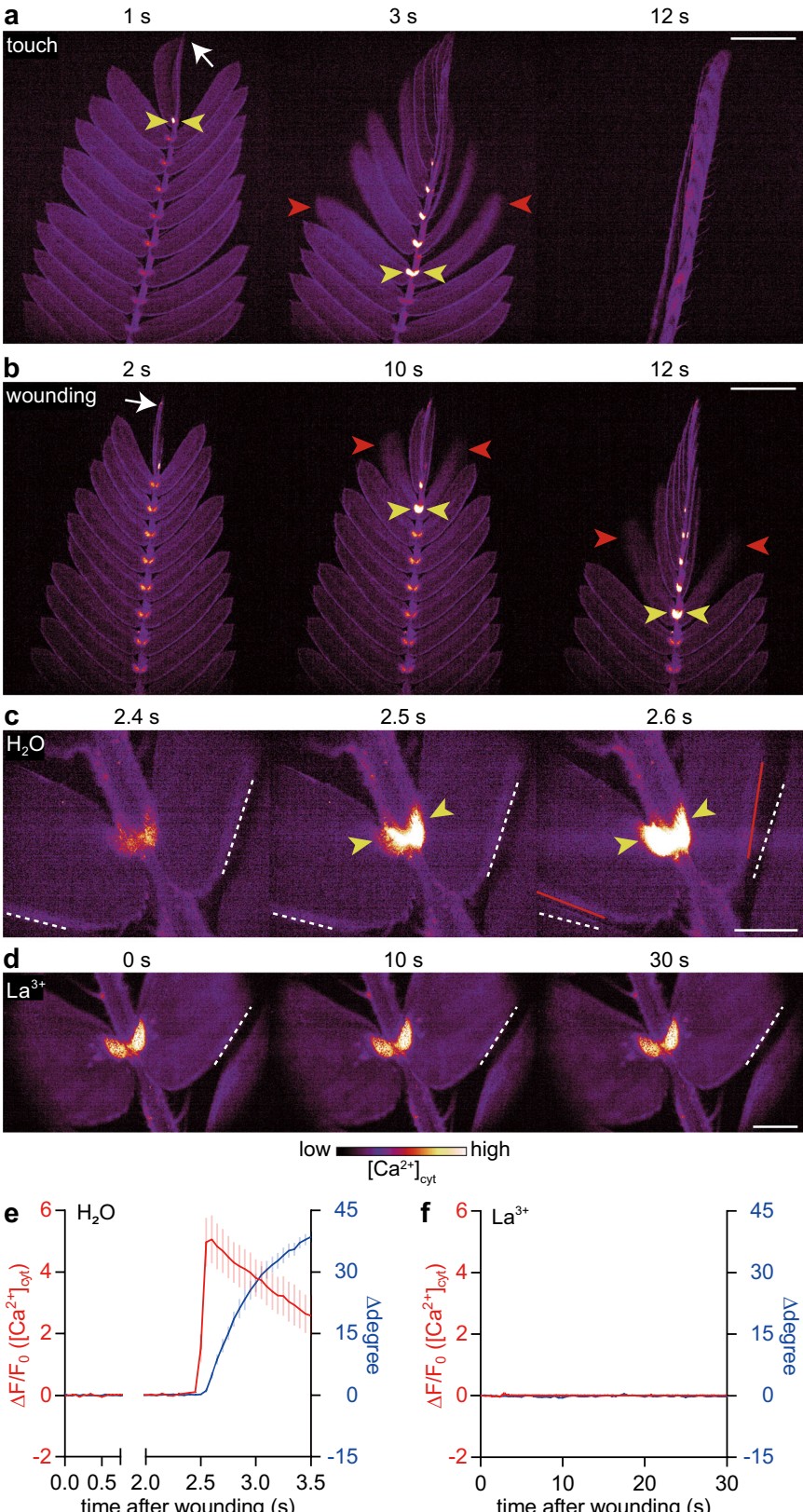

**Fig. 1 | [Ca²⁺]$_{cyt}$ increase at the pulvinus triggers rapid leaflet movement.**
**a**, **b** Touch (**a**) and wounding (**b**) (white arrows) caused [Ca²⁺]$_{cyt}$ increases at the tertiary pulvini (yellow arrowheads) and leaflet movements (red arrowheads) that propagated toward the base of the rachilla. **c**, **d** Wounding triggered [Ca²⁺]$_{cyt}$ increases at the tertiary pulvini that preceded the leaflet displacements in control (**c**) but not in La³⁺-treated leaves (**d**). Dashed white and solid red lines indicate leaflet positions before and after leaflet movements, respectively. **e**, **f** [Ca²⁺]$_{cyt}$ signatures at the tertiary pulvinus and leaflet angle in leaves pretreated with H$_2$O (**e**, $n = 5$) and 50 mM La³⁺ (**f**, $n = 7$). Mean ± SEM values are shown. Scale bars, 5 mm (**a** and **b**) or 1 mm (**c** and **d**).

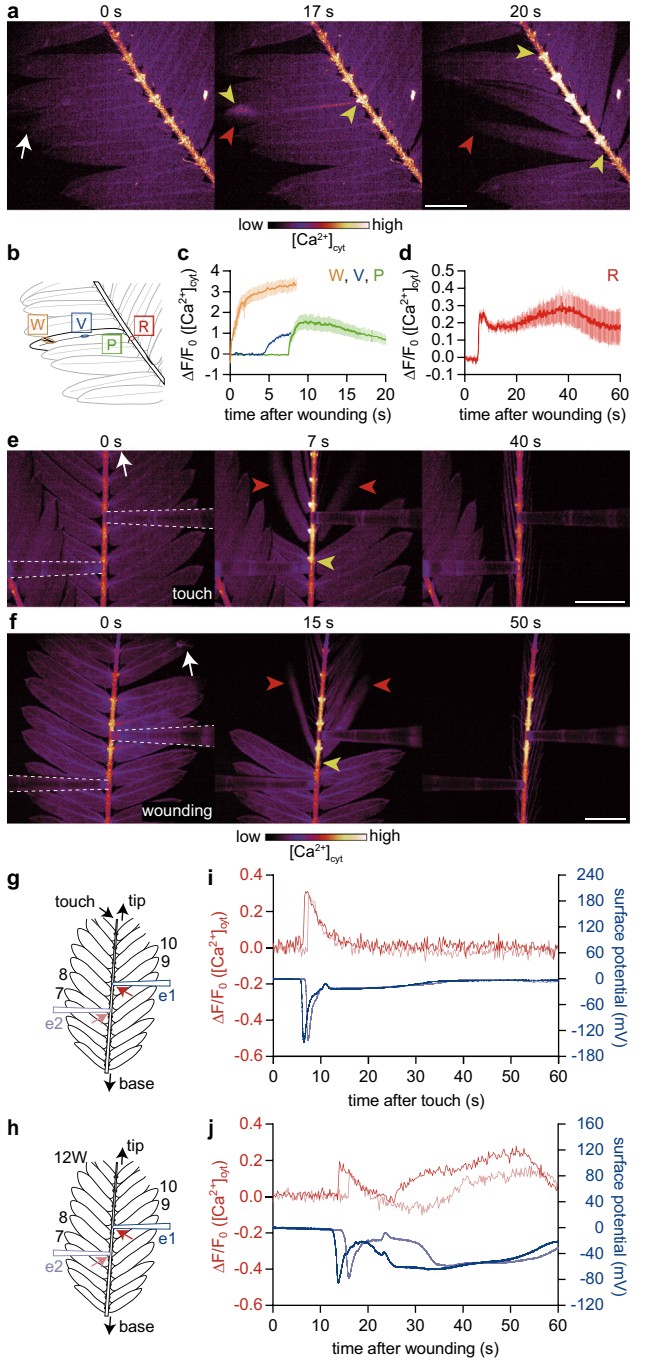

**Fig. 2 | Touch and wounding trigger long-distance rapid [Ca²⁺]$_{cyt}$ and electrical signals. a** Wounding by scissors (white arrow, 0 s) caused a [Ca²⁺]$_{cyt}$ increase that was transmitted through leaflet veins and rachillae (yellow arrowheads), leading to pulvinar movements (red arrowheads). **b** Diagram of the leaf with the regions of interest (ROIs) for [Ca²⁺]$_{cyt}$ analysis. W, wound site; V, leaflet vein; P, tertiary pulvinus; R, rachilla. **c**, **d** [Ca²⁺]$_{cyt}$ changes monitored in the W, V, P (**c**, $n = 6$), and R regions (**d**, $n = 10$). The ΔF/F$_0$ curves were terminated at the time points at which ROIs on W or V could not be traced because of leaflet movements. Mean ± SEM values are shown. **e**, **f** Simultaneous recording of [Ca²⁺]$_{cyt}$ increases (yellow arrowhead) and electrical signals and leaflet movements (red arrowhead) caused by touch (**e**) or wounding (**f**) as indicated by white arrows (0 s). **g**, **h** Electrodes (e1 and e2, blue rectangles) and ROIs (red arrows, 1 mm from the electrodes) were set on the rachilla for surface potential measurement and [Ca²⁺]$_{cyt}$ analysis, respectively. A pair of leaflets was numbered from the base of a pinna. The tip of a leaf pinna was touched by forceps (**g**), or leaflet number 12 was wounded with dissecting scissors (**h**, W). **i**, **j** Changes in [Ca²⁺]$_{cyt}$ and surface potential in response to touch (**i**) or wounding (**j**) (colors as depicted in **g** and **h**). Scale bars, 5 mm.

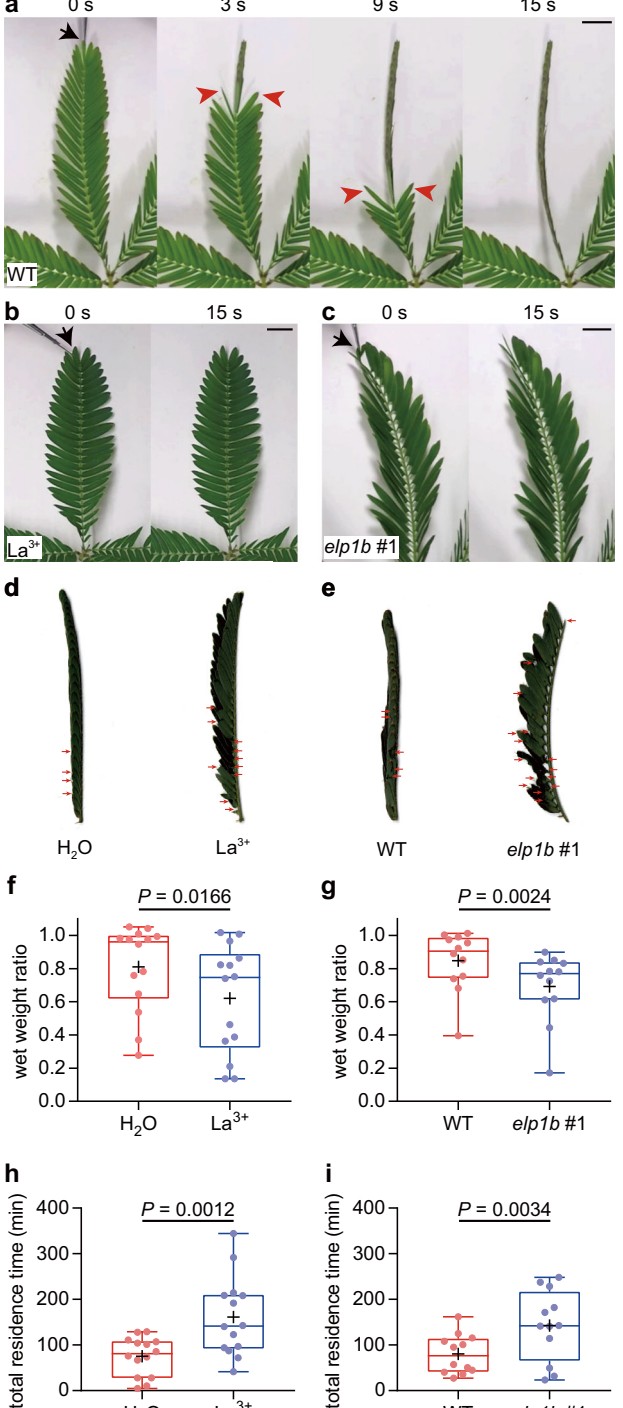

**Fig. 3 | Immotile *M. pudica* is more vulnerable to attacks by grasshoppers. a**–**c** Wounding (black arrows, 0 s) caused leaflet movements (red arrowheads) in wild-type (WT) leaves (**a**) but not in La³⁺-treated (**b**) and *elp1b1elp1b2* (*elp1b* #1, **c**) leaves. **d**, **e** Herbivory damage (red arrowheads) in H₂O- (control) and La³⁺-treated pinnae (**d**) and WT and *elp1b1elp1b2* pinnae (**e**). **f**, **g** La³⁺-treated leaves (**f**) and *elp1b1elp1b2* leaves (**g**) were more consumed by grasshoppers than control/WT leaves. **h**, **i** Total residence time of grasshoppers on the control and La³⁺-treated leaves (**h**) and the WT or *elp1b1elp1b2* leaves (**i**) $n = 14$ independent leaf pairs for **f** and **h**, and $n = 12$ independent leaf pairs for **g** and **i**. The boxes show the inter-quartile ranges, and the whiskers show the minimum and maximum values. The horizontal lines within the boxes and the plus signs indicate the medians and means, respectively. The dots represent individual data. Statistical analyses were performed using a two-tailed Wilcoxon matched-pairs signed rank test. Scale bars, 10 mm.

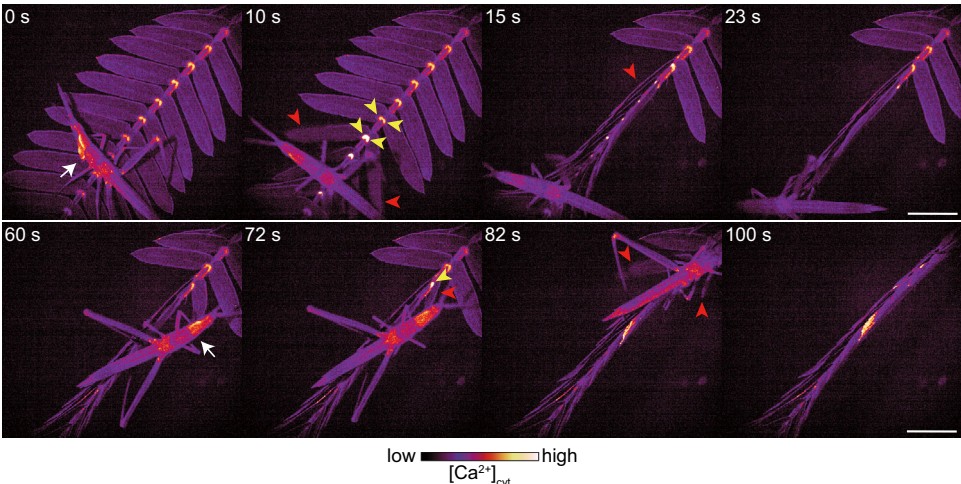

**Fig. 4 | Insect attack induces $[Ca^{2+}]_{cyt}$ increases and leaflet movements.** Feeding on a leaflet by a grasshopper (white arrows, 0 and 60 s) caused $[Ca^{2+}]_{cyt}$ increases in pulvini (yellow arrowheads) and leaflet movements (red arrowheads). Note that grasshoppers are naturally fluorescent. Scale bar, 5 mm.

deficiency by precipitating phosphate, which modulates the root system architecture in *A. thaliana*[24]. In addition, prolonged EGTA treatment alters the mechanical strength of inflorescence stems in *Paeonia lactiflora* since EGTA removes $Ca^{2+}$ from the cell wall[25]. To address these critical questions, we conducted pharmacological experiments with different concentrations and chemicals. A $La^{3+}$ solution with high $K^+$ and $Ca^{2+}$ concentrations inhibited wound-induced $[Ca^{2+}]_{cyt}$ increases at the wound sites, leaflet veins, pulvini, and rachillae and subsequent leaflet movements (Supplementary Fig. 13a–c). The dose–response curve indicated that the half maximal inhibitory concentration ($IC_{50}$) of $La^{3+}$ was 0.52 mM (Supplementary Fig. 13d). These data suggest that the inhibitory effects on $[Ca^{2+}]_{cyt}$ dynamics and leaflet movements were due to $La^{3+}$ but not to the $K^+$ deficiency and membrane depolarization. To exclude the possibility that $La^{3+}$-induced phosphate deficiency hindered $[Ca^{2+}]_{cyt}$ and electrical signals, we used verapamil, another $Ca^{2+}$ channel antagonist, which retarded $Ca^{2+}$/electrical signal propagation in the rachilla (Supplementary Fig. 14). We also reduced the period of EGTA treatment and mitigated the reduction in the $Ca^{2+}$ buffering capacity of EGTA with the pH buffer HEPES[26]. The $[Ca^{2+}]_{cyt}$ increase was limited within the wounded leaflets (Supplementary Fig. 15a, b), and the long-distance $Ca^{2+}$/electrical signal was inhibited in the rachillae of leaves treated with EGTA and HEPES for a shorter period (Supplementary Fig. 15c–f). Furthermore, we treated membrane-permeable cytosolic $Ca^{2+}$ chelator (BAPTA tetra-acetoxymethyl ester) together with EGTA to the rachillae. Despite the shortening of the incubation period to 0.5 h, wounding did not induce the long-distance propagation of the $Ca^{2+}$ and electrical signals in the rachillae (Supplementary Fig. 16). Therefore, the $Ca^{2+}$/electrical signal and leaflet movements are unlikely to be inhibited by the adverse effects of $La^{3+}$ and EGTA, supporting our model that the $[Ca^{2+}]_{cyt}$ changes coupled with the electrical signals act as the long-distance signal triggering leaf movements.

Although now strongly linked to the leaflet movements, the molecular mechanisms driving the changes in $[Ca^{2+}]_{cyt}$ and their associated electrical signals remain elusive. In plants, anions are thought to play a role in AP[27]. Indeed, the amplitude of an AP is known to be dependent on extracellular $Cl^-$[28] and $Ca^{2+}$ (Supplementary Figs. 5 and 6)[15] in *M. pudica*. We speculate that the $[Ca^{2+}]_{cyt}$ increases might activate $Ca^{2+}$-dependent $Cl^-$ channels to transmit APs, as reported in Characeae algae[29]. Since VPs are related to the chemicals released upon wounding and transported through the xylem[1,30], ligand-gated ion channels, e.g., *GLUTAMATE RECEPTOR-LIKE* family, might be involved in *M. pudica*'s VP[31]. Leaf opening was not influenced by $La^{3+}$ treatment (Supplementary Fig. 17), suggesting that the recovery process after rapid movement is regulated by signals other than $[Ca^{2+}]_{cyt}$ and AP/VP.

## Methods

### Plant material and growth conditions

Surface-sterilized seeds of *M. pudica* (an early-flowering WASE accession, Sakata Seed) were sown on germination medium [1/2× basal MS salts (Wako), 0.2% (w/v) gellan gum (Wako) and 0.05% (w/v) 2-morpholinoethanesulfonic acid (MES, Dojindo Laboratories); pH adjusted to 5.8 with KOH] in a square plate or in a mixture of culture soil (Nippi Engei Baido 1, Nihon Hiryo) and vermiculite (equal volumes) in a pot. The plates and pots of *M. pudica* were cultivated in a growth chamber (GB48, Conviron) under 12-h/12-h light/dark and 27 °C/20 °C cycles. One-week-old seedlings in the plates were used for $Ca^{2+}$ imaging of the roots and cotyledons (Supplementary Fig. 1d, e), whereas plants more than 2 months old in the pots were used for the following experiments.

### Isolation of the coding sequence of *M. pudica ELP1B* genes

Taking advantage of the intronless structure of *ELP1* coding sequences, we first isolated their partial sequences from *M. pudica* genomic DNA. The genomic DNA was obtained from *M. pudica* immature leaves using the CTAB extraction method[32]. Using KOD-Plus-Neo DNA polymerase (TOYOBO) and a pair of degenerate primers (5′-ATGGCATCATCAA GCTCHTACAATTCNCCNTGYGC-3′ and 5′-AGAAGATCAGCRTTAGC VGAATCRAGYTCYTTYTG-3′), which were designed against well-conserved N-terminal and leucine zipper-like regions of leguminous *ELP1* genes, we obtained PCR products of ~330 bp in size. These fragments were cloned into a pCR-Blunt II-TOPO vector (Thermo Fisher Scientific) and sequenced using the ABI 3130xl sequencer (Applied Biosystems) and a BigDye Terminator v3.1 Cycle Sequencing Kit (Thermo Fisher Scientific). Sequencing analysis revealed that *M. pudica* has at least 4 different sequences of *ELP1* genes, named *ELP1A1*, *ELP1A2*, *ELP1B1*, and *ELP1B2*. The doubled chromosome number found in *M. pudica*[33] indicated a recent whole-genome duplication event in this species, and we consistently observed two gene pairs with highly similar sequences. Only 8 nucleotides differed in the isolated 262-nucleotide region in *ELP1A1* and *ELP1A2*, and 4 of 262 nucleotides differed in *ELP1B1* and *ELP1B2*, indicating that these gene pairs originated from ancestral *ELP1A* and *ELP1B* by whole-genome duplication. Then, we isolated the full-length cDNA sequences of *ELP1B1* and *ELP1B2* genes by the rolling circle amplification-RACE method[34]. Briefly, total RNAs were extracted from 3-day-old seedlings or a shoot apex containing immature leaf primordia (~5 mm in length) using PureLink Plant

RNA Reagent (Thermo Fisher Scientific). The extracted RNAs were treated with DNase I and further purified using a silica membrane column (RNeasy Plant Mini Kit, Qiagen). Then, first-strand cDNA was synthesized with SuperScript III reverse transcriptase (Thermo Fisher Scientific) and phosphorylated oligo (dT)$_{12-18}$ primers (Thermo Fisher Scientific). After RNase H treatment and purification using a QIAquick PCR Purification Kit (Qiagen), the single-stranded cDNAs were circularized with CircLigase II ssDNA Ligase (Epicenter). The circularized cDNAs were purified with a QIAquick PCR Purification Kit and then subjected to rolling circle amplification with ɸ29 DNA polymerase (New England Biolabs) and phosphorothioate-modified random hexamers at 30 °C for 24 h. Using these amplified cDNAs as the template, PCR amplification was performed with the primers 5′-TATGGCTGTGTAGGAGCCATCTCC-3′ and 5′-GAGTTGACGGCGTCTTC TCTCTGG-3′. The PCR products were electrophoresed in 2% agarose gel, and several conspicuous DNA bands were excised from the gel. After purification using a FavorPrep GEL/PCR Purification Kit (FAVORGEN), the PCR fragments were phosphorylated with T4 polynucleotide kinase (New England Biolabs), purified again with a GEL/PCR Purification Kit, and then cloned into the SmaI site of the pGEM-3Zf (+) vector (Promega). We obtained the 5′ and 3′ sequences of *ELP1B1* and *ELP1B2* genes by Sanger sequencing, and the sequence information was used to design the genotyping primers described below.

## Vector construction

For the *LjUBQ1p* (*Lotus japonicas* polyubiquitin promotor[35])::*GCaMP6f* Ca$^{2+}$ biosensor construct, we first constructed a pSB11U2 vector containing *LjUBQ1p*, a multicloning site, and 35S terminator sequences in this order, together with a hygromycin resistance gene cassette. The *SacI*−*StuI* fragment of pIG121-Hm, which contained a NOS terminator and hygromycin resistance cassette, was inserted into the *SacI*-blunted *EcoRI* site of pSB11 (the resultant intermediate construct was named pSB11-NT-Hm, and we similarly indicate construct names hereafter). A 35S terminator sequence was PCR-amplified with primers (5′-CCGA GCTCGGCCATGCTAGAGTCCGCAAAAATC-3′ and 5′-AGGACGCGTAGG TCACTGGATTTTGGTTTTAGG-3′) using pK7WGF2::hCas9 as the template, digested with *SacI* and *MluI*, and then inserted into the equivalent site of pSB11-NT-Hm to replace the NOS terminator with a 35S terminator (pSB11-ST-Hm). The *LjUBQ1p* fragment was PCR-amplified with primers (5′-GCCTCGAGGAGAGAGGATTTTGAGGAAATAATTAAT TG-3′ and 5′-TCTAGACTGTAATCACATCAACAACAGATAAAT-3′) using pUB-Hyg as the template, digested with *XhoI* and *XbaI*, and then inserted into the equivalent site of pSB11-ST-Hm, generating the expression vector pSB11U2. The GCaMP6f fragment, gifted by Douglas Kim and the GENIE Project (Addgene plasmid #40755), was PCR-amplified between the *XbaI* and *BamHI* sites and inserted into the equivalent sites of the pAN19 vector. The GCaMP6f fragment was isolated by *XbaI* and *XmaI* digestion and ligated into the equivalent sites of the pSB11U2 vector.

For CRISPR/Cas9 genome editing, we constructed a pSB11CAS11 vector, which possessed expression cassettes for hCas9, guide RNA, sGFP, and hygromycin resistance genes. The entire construction procedure of this vector reflects a series of modifications for functional improvement. A 35S promoter fragment was excised from pIG121-Hm via digestion with *HindIII* and *XbaI* and then inserted into the equivalent site of pSB11-NT-Hm (pSB11S). A DNA fragment containing EGFP-hCas9 and the 35S terminator was PCR-amplified with primers (5′-AC TCTAGACACCATGGTGAGCAAGGGCGAGGAG-3′ and 5′-AGGACGCG TCAGGTACCAAGGTCACTGGATTTTGGTTTTAGG-3′) using pK7WGF2:: hCas9 as the template, digested with *XbaI* and *MluI*, and then inserted into the equivalent site of pSB11S (pSB11CAS). An *Arabidopsis* U6 promoter fragment was PCR-amplified with primers (5′-CAAGGTCTC GGTACCGGAGTGATCAAAAGTCCCAC-3′ and 5′-AATGGTCTCCAATCG CTATGTCGACTCTATC-3′) using pICH86966::AtU6p::sgRNA PDS as the

template and then digested with *BsaI*. Similarly, a DNA fragment containing the guide RNA backbone and poly T terminator was PCR-amplified with primers (5′-TGAGGTCTCAGTTTTAGAGCTAGAAATAGC AAG-3′ and 5′-GAAGGTCTCACGCGTAAAAAAAGCACCGACTCGGTG-3′) using pICH86966::AtU6p::sgRNA PDS as the template and then digested with *BsaI*. Double-stranded DNA oligo containing a target sequence for *M. pudica ELP1A* genes was prepared by annealing two synthesized oligonucleotides (5′-GATTGCAAACGTCCATAAAATATT-3′ and 5′-AAACAATATTTTATGGACGTTTGC-3′). These three fragments were introduced into the *Acc65I*−*MluI* site of pSB11CAS (pSB11CAS-ELP1A) through 4-fragment ligation to reconstruct a functional guide RNA expression cassette. This initial CRISPR/Cas9 construct was further subjected to two serial modifications. The first modification was intended to replace the 35 S promoter upstream of EGFP-hCas9 with *LjUBQ1p* and also to rearrange the position and orientation of the expression cassettes. To this end, a *LjUBQ1p* fragment was excised from pSB11U2 via digestion with *XhoI* and *XbaI*. A DNA fragment containing EGFP-hCas9 and the 35S terminator was excised from pSB11CAS-ELP1A via digestion with *XbaI* and *MluI*. A DNA fragment containing the guide RNA cassette was PCR-amplified with primers (5′-TCCTCGAGTTGCTAGCGGAGTGATCAAAAGTCCCAC-3′ and 5′-GCAA GCTTGGACCGGTAAAAAAAGCACCGACTCGGTG-3′) using pSB11CAS-ELP1A as the template and then digested with *XhoI* and *HindIII*. These three fragments were introduced into the *HindIII*−*MluI* site of pSB11-ST-Hm via 4-fragment ligation (pSB11CAS10-ELP1A). As the fluorescence intensity of EGFP-hCas9 fusion proteins was insufficient for use as a visual selection marker, we further intended to assign hCas9 and GFP to two separate expression cassettes. To this end, a DNA fragment containing a sGFP expression cassette was PCR-amplified with primers (5′-TTACCGGTGCATGCCTGCAGGTCCCCAG-3′ and 5′-AATAAGCTT CCCGATCTAGTAACATAGATGACACC-3′) using pIF121-Hm[32] as the template and then digested with *AgeI* and *HindIII*. A DNA fragment containing the guide RNA cassette and *LjUBQ1p* was excised from pSB11CAS10-ELP1A via digestion with *AgeI* and *NcoI*. Similarly, a DNA fragment containing hCas9 and the 35 S terminator was excised from pSB11CAS10-ELP1A via digestion with *NcoI* and *MluI*. These three fragments were introduced into the *HindIII*−*MluI* site of pSB11-ST-Hm via 4-fragment ligation (pSB11CAS11-ELP1A). Finally, we replaced the pre-existing target sequence with that of *ELP1B* genes. To this end, double-stranded DNA oligo containing the *ELP1B* target sequence was prepared with oligonucleotides (5′-GATTGGTGAGGAAGGAGCT CATTG-3′ and 5′-AAACCAATGAGCTCCTTCCTCACC-3′). This target sequence and adjacent PAM sequence were completely conserved in *ELP1B1* and *ELP1B2* genes, and thus, the single target sequence could be expected to recognize both genes simultaneously. A U6 promoter fragment was PCR-amplified with primers (5′-CAAGGTCTCGCTAGCG GAGTGATCAAAAGTCCCAC-3′ and 5′-AATGGTCTCCAATCGCTATGTC GACTCTATC-3′) using pSB11CAS11-ELP1A as the template and then digested with *BsaI*. Similarly, a DNA fragment containing the guide RNA backbone and poly T terminator was PCR-amplified with primers (5′-TGAGGTCTCAGTTTTAGAGCTAGAAATAGCAAG-3′ and 5′-GAAGGT CTCACCGGTAAAAAAAGCACCGACTCGGTG-3′) using pSB11CAS11-ELP1A as the template, and then digested with *BsaI*. These three fragments were incorporated into the *AgeI*−*NheI* site of pSB11CAS11-ELP1A to replace the guide RNA cassette. The resultant vector pSB11CAS11-ELP1B was used for the subsequent knockout experiment.

Next, pSB11U2-GCaMP6f or pSB11CAS11-ELP1B were electroporated into *Agrobacterium* LBA4404 harboring a pSB1 acceptor vector to produce hybrid pSB111U2-GCaMP6f and pSB111CAS11-ELP1B vectors through homologous recombination[36]. The transgenic plants were produced using the *Agrobacterium*-mediated method as described previously[32]. Briefly, cotyledonary node explants were cocultivated with the *Agrobacterium* suspension for 3 days and then subjected to drug selection on a selection medium [1/2× basal MS salts, 2% (w/v) sucrose, 1× Gamborg's vitamins (Sigma-Aldrich), 0.5 μg/ml

6-benzylaminopurine (Sigma-Aldrich), 0.3% gellan gum, 15 μg/ml hygromycin B (Sigma-Aldrich) and 150 μg/ml cefotaxime sodium salt (Sanofi K.K.) at pH 5.8] for about 2 months. GFP-positive callus was surgically excised from the explant and then the shoot and root were induced using a shoot induction medium (1/2× MS, 2% (w/v) sucrose, 1× Gamborg's vitamins, 0.5 μg/ml 6-benzylaminopurine and 0.3% gellan gum at pH 5.8) and sterilized irrigated vermiculite (Fujimi Engei), respectively[32].

## Mutation analysis of CRISPR/Cas9-mediated mutants

Eight independent transgenic $T_0$ plants were established for pSB111CAS11-ELP1B. After transfer into soil and subsequent plant growth, an immature leaf was excised and used for genomic DNA extraction using a DNeasy Plant Mini Kit (Qiagen). Genomic DNA fragments of *ELP1B1* were PCR-amplified with a common ELP1B forward primer (5′-TGCAAATTCCTGAGGAGGAAGTGCA-3′) and a specific ELP1B1 reverse primer (5′-TAGTCCCTGATCTTGCCCTCTG*A*T-3′; asterisks indicate the two phosphorothioate modifications to prevent unwanted degradation of the primer by the proofreading activity of KOD-Plus-Neo DNA polymerase). Similarly, genomic DNA fragments of *ELP1B2* were PCR-amplified with the common forward primer and a specific ELP1B2 reverse primer (5′-TAGTCCCTGATCTTGCCCTCTG*A*G-3′). After purification with a FavorPrep GEL/PCR Purification Kit, the DNA fragments were directly sequenced in both directions using an ABI 3130xl sequencer, a BigDye Terminator v3.1 Cycle Sequencing Kit, and the forward and reverse PCR primers. The electropherograms of the DNA sequencing, in which two different peaks were simultaneously detected because of the different sizes of indels in the two mutated alleles, were optically decomposed to each single sequence. Sequence analysis revealed various patterns of nucleotide insertions and deletions into the target genes between *elp1b1elp1b2* line 1 and 8 $T_0$ plants (Supplementary Figs. 9 and 10a).

## High-speed and wide-field Ca²⁺ imaging

For leaf imaging, the leaves were severed at the petiole bases, and the leaf petioles were inserted into water-filled 15-ml tubes through pores on the caps. The leaves were fixed with Parafilm and recovered at around 25 °C under white LED light. When the leaflets were opened, the tube with the leaf specimen was carefully transferred onto the stage of a fluorescent microscope. For Ca²⁺ imaging using a grasshopper, the leaf was floated onto the stage.

The transgenic *M. pudica* was imaged in real time with a motorized fluorescence stereomicroscope (SMZ25, Nikon) equipped with a 1× objective lens (NA = 0.156, P2-SHR PLAN Apo, Nikon) and a sCMOS camera (ORCA-Flash4.0 V2, Hamamatsu Photonics)[19] or a motorized macro zoom microscope (Axio Zoom.V16, Zeiss) equipped with a 1× objective lens (NA = 0.25, PlanNeoFluar Z, Zeiss), an optical beam splitting system (W-VIEW GEMINI-2C, Hamamatsu Photonics), and a sCMOS camera (ORCA-Flash4.0 V3, Hamamatsu Photonics). The GCaMP6f Ca²⁺ indicator was excited with a mercury lamp, 470/40-nm excitation filter, and 500-nm dichroic mirror (P2-EFL GFP-B, Nikon) under the SMZ25 microscope and with a mercury lamp, an excitation filter (59012x, Chroma), and a dichroic mirror (T562lpxr, Chroma) under an Axio Zoom.V16 microscope. The fluorescence signals passing through a 535/50-nm filter in SMZ25 or a 519/26-nm filter in Axio Zoom.V16 were acquired by the cameras at arbitrary time intervals (3–20 frames/s) using imaging software [NIS-Elements Advanced Research, Nikon, or ZEN pro (blue edition), Zeiss]. During imaging, the image sequences were processed in a 4 × 4 binning mode to allow rapid imaging. Figure 2a was corrected by non-linear adjustment (gamma value = 1.50).

Upon fluorescence imaging, *M. pudica* leaves were stimulated with a touch by tweezers (or a micropipette tip), wounding by dissecting scissors, or grasshopper (*Acrida cinerea*) feeding.

Several ROIs were analyzed over time using NIS-Elements and ZEN pro imaging software (for ROI positions, see Fig. 2b or Supplementary Fig. 3c). Background noise was subtracted from the GCaMP6f signals on the leaves over time. The fractional fluorescence changes were calculated according to the equation $\Delta F/F_0 = (F - F_0)/F_0$, where $F$ denotes GCaMP6f fluorescence at a certain time and $F_0$ denotes the averaged baseline fluorescence defined as the average of $F$ over -1 s without cytosolic Ca²⁺ ($[Ca^{2+}]_{cyt}$) changes.

## Simultaneous measurement of Ca²⁺ and electrical signals

KCl, CaCl₂ (both from Wako), and MES were dissolved in water to prepare stock solutions (100× for MES, 1000× for KCl and CaCl₂). Extracellular fluid [ECF; 0.1 mM KCl, 1 mM CaCl₂, and 1 mM MES; pH adjusted to 5.8 with Tris(hydroxymethyl)aminomethane (Rikaken)] was made by diluting the stock solutions. The leaf specimens were treated as described in the Ca²⁺ imaging section using ECF instead of water.

Recording electrodes were prepared according to Sibaoka[37] with the following modifications: Ag/AgCl wires of 0.2 mm in diameter were inserted into truncated micropipette tips, and the tips were fixed to electrode holders and filled with 10 mM (or in some cases, 1 mM) KCl solution. A handmade electrode or, in some cases, an Ag/AgCl-type pellet electrode (1-HLA-003, Inter Medical), was used as a reference electrode. The handmade reference electrode was prepared by inserting an Ag/AgCl wire of 0.5 mm in diameter into a truncated micropipette tip that was filled with 10 mM KCl solidified by 0.5% (w/v) agarose (Rikaken). Two operational amplifiers (Axopatch 200 A, Axon Instruments), two headstage amplifiers (CV-201A headstage, Axon Instruments), a digitizer (Digidata 1322 A, Axon Instruments), and electrophysiology data acquisition software (Clampex 9.2, Axon Instruments) were used to detect surface potential changes. Simultaneous measurements were performed under the SMZ25 microscope in a Faraday cage. The leaf sample was placed upside down on the stage and exposed to excitation light. The reference electrode was inserted into the bathing fluid, and the two recording electrodes were placed on a rachilla (for recording electrode placements, see Fig. 2g, h and Supplementary Fig. 4a). The GCaMP6f fluorescence was acquired every 200 ms, and the surface potential changes were sampled at 5 kHz and low-pass filtered at 1 kHz. The fluorescence signals were analyzed as described previously (for ROI positions, see Fig. 2g, h and Supplementary Fig. 4a). The surface potential changes (ΔV) were calculated using the equation $\Delta V = V - V_0$, where $V$ denotes the potential difference (PD) between the recording and the reference electrodes at a certain time and $V_0$ denotes the averaged baseline PD over 1 s with no electrical changes.

## Pharmacological treatments

LaCl₃·7H₂O (Wako) was dissolved in ECF (Supplementary Fig. 5) in high K⁺ and Ca²⁺ solution [20 mM KNO₃, 10 mM Ca(NO₃)₂, 1 mM MES; pH adjusted to 5.8 with Tris(hydroxymethyl)aminomethane; Supplementary Fig. 13] or in water for other experiments at a final concentration of 0.1–50 mM. EGTA (Dojindo Laboratories) was dissolved in water (Supplementary Figs. 3b–f and 6) or 10 mM HEPES (Supplementary Fig. 15) to make a 50-mM solution, and the pH was adjusted to 7.0 with KOH and HCl. Verapamil hydrochloride (Sigma-Aldrich) was dissolved in ECF at a final concentration of 2 mM. The petioles of the leaf specimens were dipped into and treated with these solutions in 15-ml tubes for 2–3 h for La³⁺ treatment, 3–5 h for EGTA treatment, or 5 h for verapamil treatment under white LED light. The leaves were then dipped in a control solution or 1 mM KCl for EGTA-treated leaves (Supplementary Fig. 6) to stop the pharmacological treatments and were allowed to recover for at least 1 h under white LED light.

BAPTA tetraacetoxymethyl ester (BAPTA-AM, Tokyo Chemical Industry) was dissolved in DMSO to make a 100-mM stock solution. The stock solution was diluted 100 times using 10 mM HEPES solution

with 50 mM EGTA, and the pH was adjusted to 7.0 using KOH and HCl. Pinnae were isolated from the leaves, and basal 7 leaflet pairs were removed from the base of the isolated pinnae to insert the pinnae into the solution. The isolated pinnae were inserted into 1.5-ml tubes filled with water through holes on the lids, fixed with Parafilm to the tubes, and placed in an incubator (LU-113, ESPEC) at 25 °C overnight. The pinnae were transferred into PCR tubes with 0.3 ml of EGTA and BAPTA-AM, fixed to the tubes with Parafilm, and treated with the chelators under white LED light in the incubator at 25 °C for 0.5 h. The treated pinnae were then put via holes in the lids into 1.5-ml tubes filled with 10 mM HEPES (pH was adjusted to 7.0 with KOH and HCl), fixed to the tubes with Parafilm, and maintained at ~25 °C for more than 1 h under white LED illumination. Subsequently, the $Ca^{2+}$ and electrical signals were measured (for positions of ROIs and recording electrodes, see Supplementary Fig. 16a).

### Videography and photography
Movies of leaflet movements were acquired with a video camera (HERO5 BLACK, GoPro) or the aforementioned imaging systems, and the leaflet angle changes were estimated using NIS-Elements imaging software. Photographs of various tissues were collected by a stereomicroscope (Leica S9D, Leica Microsystems) equipped with a camera (Leica MC190 HD, Leica Microsystems) and processed using microscope software (Leica Application Suite, Leica Microsystems). The images acquired using a Leica S9D microscope (Supplementary Fig. 10d–f) were corrected by non-linear adjustment (gamma value = 0.60).

### Feeding assay
WT and *elp1b1elp1b2* leaves with 4 pinnae were cut at the petiole bases. A pair of control and 50 mM $La^{3+}$-treated leaves or WT and *elp1b1elp1b2* leaves was placed into a 50-ml glass bin filled with water. The leaves were fixed to the bins with a piece of Parafilm. Several bins containing leaves were placed in a plastic container ($53 \times 39 \times 32$ cm$^3$), and then the leaves were recovered at room temperature under white LED light. When the WT (control) leaves were expanded, 3–7 adult grasshoppers (*Atractomorpha lata*, ~4 cm in length) or 2 caterpillars (*Helicoverpa armigera*, ~2.5 cm in length) per bin, which had been fasted for ~1 day, were put in the container. The assays were monitored with a HERO5 BLACK video camera for 3 h for the grasshoppers and 2 h for the caterpillars. The assays were repeated with off-food intervals (approximately 1 day). In every assay, the fresh leaves were weighed on an electronic balance (GR-202, A&D Company) before and after the experiments, and the ratio of these values was calculated. Thereafter, the pinnae were detached from the leaves using dissecting scissors and scanned by a photocopier (EP-M570T, Seiko Epson). The total residence time of the grasshoppers on a leaf was determined from the movie data. The transgenic leaves expressing *LjUBQ1p::GCaMP6f* were used as the WT leaves in these experiments.

### Statistical analysis
The time point of the $[Ca^{2+}]_{cyt}$ increase was analyzed by *t*-test using the criterion of a rise to $3\times$ SD above $F_0$ as an indicator of a detectable increase in the GCaMP6f signal, and the speed of the $[Ca^{2+}]_{cyt}$ signal was then calculated by dividing the distance between two ROIs by the interval between the calculated time points at the two ROIs using Excel (Microsoft)[19]. The timing of the detectable leaflet angle change was similarly determined. The inter-electrode distances were obtained from the imaging data, and the peak times of the surface potential changes were determined using electrophysiology analysis software (Clampfit, Axon Instruments). The action potential velocities were calculated using Excel by dividing the inter-electrode distances by the time intervals between the peak-to-peak time points.

We used GraphPad Prism (GraphPad Software) for the following statistical analyses. The velocities of the $[Ca^{2+}]_{cyt}$ increases and surface

potential changes, the wet weight ratio of the leaves before and after the feeding assays, and the total residence time of insects on the leaves were analyzed by a two-tailed Wilcoxon matched-pairs signed rank test. $IC_{50}$ for the inhibitory effects of $La^{3+}$ was estimated by fitting a sigmoidal dose–response curve. Mean ± SEM values were represented in the main text and Supplementary Table 1 for the velocities of signal propagations and the total residence time of insects on the leaves.

### Reporting summary
Further information on research design is available in the Nature Research Reporting Summary linked to this article.

## Data availability
The data that support the findings of this study are available from the corresponding author upon reasonable request. Plasmid sequence data can be found in GenBank with the accession numbers pSB1 (AB027255) and pSB11 (AB027256). The GCaMP6f sequence is available at Addgene (#40755). *ELP1B1* and *ELP1B2* sequences are deposited in DNA Data Bank of Japan with the accession numbers *ELP1B1* (LC731322) and *ELP1B2* (LC731323). Source data are provided with this paper.

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

## Acknowledgements

We thank Simon Gilroy for critical reading of the manuscript. This research was supported by grants from KAKENHI (21J23365 to T.H., 16K07411 and 19K06715 to H.M., 17H06390 to M.H., 21H04978 to H.M., M.T., and M.H. and 18H05491 to M.T.), JST PRESTO to H.M., and JST A-STEP to M.T.

## Author contributions

T.H., H.M., M.H., and M.T. designed the study. H.M. produced the transgenic plants and performed genotyping. T.H. and T.M. performed $Ca^{2+}$ imaging, electrophysiological measurements and feeding assays, and analyzed all the data. T.H. and M.T. wrote the manuscript. All authors discussed the results and contributed to the manuscript.

## Competing interests

The authors declare no competing interests.

## Additional information

**Correspondence and requests** for materials should be addressed to Mitsuyasu Hasebe or Masatsugu Toyota.

