## [Peer Review File · Nature Communications]

Calcium-mediated rapid movements defend against herbivorous insects in *Mimosa pudica*.Reviewer #1 (Remarks to the Author):

This article by Hagihara et al. has provided mechanistic insight into how the *Mimosa pudica* plant that moves its leaves in reaction to external stimuli, such as touch or injury, defends against herbivorous insects. Previously, it was known that touching a specific *Mimosa pudica* leaf generates an action potential that induces loss of turgor pressure, rapid movement, and leaf folding, a behavior that was speculated to startle insects. Hagihara et al. has now shown that such responses are generated by rapid changes in calcium concentration and electrical signals that propagate from the leaflet to the pulvinar cell. Further, Hagihara et al. has provided evidence that a remarkable reduction in calcium influx by pharmacological modulators, as well as CRISPR-Cas9 genome editing technology, can block leaf movements and confer vulnerability to herbivorous insects. The experiments and conclusions are in favor of their proposed hypothesis. However, I have serious concerns regarding the pharmacological approaches used in this study.

Major points:

1) Ca^{2+} has been considered a long-distance defense signaling molecule. To determine if Ca^{2+} is required in plant-herbivore interactions, researchers often employ the Ca^{2+} signaling disruption approach. This strategy involves either restricting Ca^{2+} entrance or altering the quantity of accessible Ca^{2+} .

(A) Inhibition of Ca^{2+} entry can be achieved by using either 1) Non-Selective Ca^{2+} channel blockers such as Ca^{2+} channel blockers such as lanthanum (La^{3+}), gadolinium (Gd^{3+}) and Ruthenium Red or, 2) L-Type Calcium Channel Antagonists (verapamil, nifedipine, bepridil), iGluR/GLR Antagonists ((DNQX, CNQX, MNQX).

In this study, Hagihara et al. have used La^{3+} to inhibit intracellular Ca^{2+} increase, electrical signaling, and rapid movement in *M. pudica*. While the results clearly supported the involvement of cytosolic Ca^{2+} in plant rapid movement, the use of La^{3+} is often associated with several side effects. La^{3+} has been known to induce phosphate deficiency; La^{3+} concentration greater than 100 μM caused changes in root architecture in the *Arabidopsis* plant (Ruiz-Herrera et al., *Plant Soil* 2012, 353, 231–247). When La^{3+} is exposed to low phosphate concentrations, it can create insoluble salts. Moreover, the activity of La^{3+} may be affected by the concentration of Ca^{2+} in the medium (Corzo et al., *J. Gen. Microbiol.* 1992, 138, 1791–1795). It has also been shown that whereas La^{3+} inhibits calcium influx in the short term in response to low K^{+} , 25 μM La^{3+} amplifies the long-term secondary Ca^{2+} signal caused by K^{+} deprivation in *Arabidopsis* roots. (Behera et al., *New Phytol.* 2017, 213, 739–750). Hagihara et al. have used 50 mM La^{3+} without K^{+} to inhibit leaf movement, electrical signaling in their experiments. Based on the previous observation, the authors need to exclude these La^{3+} side effects from their data and how they may affect the results. The authors could test the effect of lower concentrations of La^{3+} and not just 50 mM concentration to determine if there is a threshold level for La^{3+} inhibitory action.

(B) The alteration of the amount of available Ca^{2+} can be achieved by chelation with Ca^{2+} chelators, such as EGTA, EDTA, and BAPTA, which bind to Ca^{2+} ions with high affinity in a selective, reversible manner. In this study, the authors have used 50 mM EGTA to alter $[\text{Ca}^{2+}]_{\text{cyt}}$ increase and movement in response to wounding in *M. pudica*. While EGTA is a strong Ca^{2+} chelator, it does have certain obvious drawbacks that should be considered while designing an experiment. One feature of Ca^{2+} chelators is that they are not only in equilibrium with Ca^{2+} ions, but also with H^{+} , as Ca^{2+} binding to a chelator typically results in the release of H^{+} (reducing the pH). The converse is likewise true for pH-sensitive chelators such as EGTA, where a change in pH significantly decreases the Ca^{2+} buffering capacity. Thus, it is critical to utilize a pH buffer in conjunction with the Ca^{2+} chelator in order to minimize the adverse effects of pH variations on the chelator's capacity to buffer Ca^{2+} ions. Additionally, prolonged EGTA treatment may alter the cell wall characteristics by decreasing the amount of Ca^{2+} -bound pectin (Patton et al. *Cell Calcium* 2004, 35, 427–431). Notably, effective Ca^{2+} chelation with EGTA needs a pH of 8.0. Hagihara et al. have used 50 mM EGTA without any pH buffer for 5 hours. Therefore, it would be ideal to test the effect of EGTA prepared in a pH buffer, incubated for a shorter time period, and observe that the obtained result is still consistent.

3) For the quantification of Ca^{2+} imaging experiments, the authors have used 3-5 sample N values in many experiments. To bolster the finding, it would be preferable to conduct these

experiments with a more number of sample N values to ensure that the results are more repeatable.

Minor points:

- 1) Inconsistency in the depiction of the Y-axis "time after wounding" timing: several graphs missing the 0-time point, for example, 1e, 2g, Extended Data Fig. 4b, Fig.6e.f,g,h, Fig. 7e.f,g,h.
- 2) Gap missing in the Figure legend: Fig 1, n=5, n=7
- 3) Gap missing Line 77: n=10; Line 119: n= 14; line 132: n= 12

Reviewer #2 (Remarks to the Author):

The manuscript by Hagihara and co-workers deals with a fascinating topic, the underlying signalling processes leading to the extremely fast leaf(let) movements in *Mimosa pudica* upon touch and wounding, which is almost unique in plant kingdom. Further, they addressed the longstanding question of the ecological relevance of this movement. Using a combination of reporter gene-based (GCaMP6f) cytosolic Ca²⁺ imaging, electrophysiology, pharmacology, and CRISPR/Cas9 generated immotile mutants (elp1b1elp1b2) the authors provide a wide range of convincing results.

The study shows the involvement of both electrical signalling, i.e. action potential and variation potential, and transient cytosolic Ca²⁺ level increase in the wound induced leaf movements. Both signals propagate fast (few mm/sec) along the veins acro- and basipetally. Employing grasshoppers and herbivorous lepidopteran larvae it was demonstrated that immotile *M. pudica* plants are more susceptible to insect feeding compared with WT plants. The authors suggest that the presentation of trichomes after leaflet movements or leaves touching the insect body are responsible for the protection. Maybe the sudden strong movement of the leaf surface scares the insects in addition?

The MS text, all figures and videos are presented very well; conclusions are justified by the results. Remaining open questions such as the nature of the molecular mechanisms driving the Ca²⁺ transients and/or the electrical signals and the channels that may be involved have already been formulated in the conclusions. Expecting answers to these questions in the present work would go far beyond the aim of the study.

To be honest, it was really fun to read this fantastic work.

Reviewer #3 (Remarks to the Author):

The Editor, Nature communications March 24, 2022

Re: Reviewing the manuscript: Calcium-mediated rapid movements defend against herbivorous insects in *Mimosa pudica*.

Dear Editor,

I was very happy to review this manuscript, thanks for the opportunity. This paper is a class for itself. Very very well done. It was a great pleasure to carefully examine the videos and figures. With all my love to my original contribution to defensive plant movement, the paper you asked me to review is much better than anything published in this subject since Darwin's time! In general, there are only a few papers that showed actual defense from herbivory by plant movement.

I have no conflict of interests, and I don't know the authors.

(1) The most critical result of that paper is that the authors showed by blocking the leaflet movements that insects attacked the disabled leaves much more than the ones that can fold their leaflets. This is a wonderful demonstration of the defensive role of leaflet folding, arriving after many decades of theoretical discussions (including mine).

(2) The use of CRISPER for disabling plant movement will show the way to many other scientists not only in arresting leaflet movement, but in many other types of functions. The use of chemical blockers to arrest leaflet movement is also impressive and very innovative concerning anti-herbivory defense. This paper opens new avenues. I hope that the authors will do the same with *Desmodium motorium* (the Indian telegraph plant) that I discussed theoretically.

(3) From the above, it is clear that the paper is first class in originality. It is exciting to read it, and to think about many potential studies that can and will follow, and not only in defensive plant movements. They are the first, and I cannot compare it to other papers in defensive plant biology.

(4) The work supports the conclusions. The methods are excellent/brilliant and can be reproduced. There is nothing that is not OK, and I have only several small suggestions about the wording that I will list below.

(5) As usual, my name Simcha Lev-Yadun should be revealed to the authors. I suggest changing several words.

(I) I suggest to change line 113 to:leaves that did not respond to wounding or touch.....

(II) I suggest to change line 115 to:more than on the control leaves.....treated leaves lost 38.0%...

(III) I suggest to change in line 133 the word "Thus" with "Since"

(IV) I suggest to change line 151 to:of the phytohormones ethylene and jasmonate.....

Sincerely,

Dr. Simcha Lev-Yadun, Professor

Reviewer #4 (Remarks to the Author):

The movements of *M. Pudica* continue to fascinate plant biologists. Much early literature has investigated the role of electrical signaling and Ca^{2+} in leaf movements in this plant. The current manuscript by Hagihara et al. applies (for the first time to my knowledge) the tools of molecular genetics to investigate the roles of stimulus-induced leaf movements in *M. pudica*. In these experiments, the authors generated *M. pudica* plants that lack functional pulvini and therefore lack the ability to move when touched or wounded. To do that they cleverly targeted and mutated LOB domain transcription factors known to be necessary for pulvinus formation in other members of the Fabaceae. This revealed that the leaves of these plants were eaten more quickly by insects than those of the wild type. This finding is perhaps the most innovative part of this manuscript.

In other experiments Hagihara et al. investigated the role of Ca^{2+} signaling in leaf movement. To do this *M. pudica* was transformed (again for the first time to my knowledge) with *GCaMP6f*. The overall findings are that pulvinar Ca^{2+} increases occur prior to leaflet movements. The new Ca^{2+} data go well beyond what has been published previous with non-transgenic approaches since they have high temporal resolution. However, data from parallel pharmacological experiment seem to be over-interpreted. This leads to statements that imply causality rather than correlation. Also, some important background and methods details are lacking. Overall, the manuscript is clearly

written, but it gives the impression of being a mixture of different objectives (defense, calcium signaling, wounding, cold treatments and pharmacological experiments) that are not always linked well in the text. The videos are spectacular and are a useful addition-especially those showing simultaneous Ca²⁺ and electrical recordings.

Major points

1. For each CRISPR-Cas9 line the authors show two mutated sequences (Extended data Figure 10). The authors cite reference 26 in the Methods section, but are they sure (given the common diversity of ploidy levels in *Mimosa* species) that the plants they used were tetraploid? The authors should state clearly how many ELP1B genes there are in the plants they used and whether all copies of each gene were mutated.

2. Key experiments use EGTA or La³⁺ treatments to block Ca²⁺ signaling. These treatments block leaf movements and cytosolic Ca²⁺ increases. The authors conclude (lines 66, 67) that cytosolic calcium increases in the pulvinus cause rapid movements. However, correlation is not causation. La³⁺ (50 mM) or EGTA treatments are expected to block much of a cell's function. The authors can only conclude that blocking cytosolic calcium signaling correlates with inhibiting leaf movements.

3. Lines 92-100. The cold treatments don't seem to fit in with the rest of the paper which concentrates on wounding and herbivory. If the cold treatment data is left in it should be better integrated in the text. For example, ED Figure 3 uses cold water treatments to separate APs and VPs. Do leaf movements still take place under the authors conditions? I think this is already known and the authors should check reference 4. If not, they should relate their data for bimodal Ca²⁺ peaks and electrical signals seen after wounding (Figure 2) to the cold water experiments. Is only the AP necessary for leaf movement under their conditions? How are the cold treatments related to herbivory and defense?

Other points

Line 54 'To determine how' ... Here, since the authors do not generate mechanistic insights, it would be more realistic to write 'To determine whether...'

Lines 96, 97. The text, which relates to Extended Data Figure 5c and 5d, is not clear to me. Figure 5c implies that consecutive cold treatments on the same sample plant were used to estimate refractory periods. In Figure 5c, after an initial cold-induced AP the next event is produced after 2, plus 3 min, plus 4 min, plus 5.3 min (= 14.3 min). It is usual to establish refractory periods with a single time interval between the first and last cold treatment.

Lines 110, 111 The authors cite three references for the putative antiherbivore defense role of leaf movements in *M. pudica*. Please note that this idea greatly pre-dates the references cited. The authors should cite reference 2 (Bose, 1926) here.

Fig. 2c The number of replicates should be given. There is no visible error envelope for the leaflet vein data. Is this because few replicates were used?

Fig. 2d The number of replicates should be given.

Extended data Fig. 10 The authors should give the exact positions of the mutations relative to the predicted transcription start sites.

Extended data Fig. 13 This figure does not seem to fit in with any other results or observations and could be removed. Hairs on petioles seem irrelevant here since the authors look at damage to leaflet on rachillae. The *elp1b* mutants are still likely to be hairy and, in any case many other factors could determine the defenses of leaves.

In the Methods section the text needs improvement:
Line 337 'was a bit complicated' should be removed

Line 391 'threes'

Line 454 'were recovered in ECF'. What does 'recovered' mean here?

Reply to Reviewers

We very much thank the reviewers for their encouraging comments and the thoughtful reviews that have helped us significantly improve our manuscript. We have addressed all the points raised and the specific reply for each query is detailed below:

Reviewer #1:

This article by Hagihara et al. has provided mechanistic insight into how the Mimosa pudica plant that moves its leaves in reaction to external stimuli, such as touch or injury, defends against herbivorous insects. Previously, it was known that touching a specific Mimosa pudica leaf generates an action potential that induces loss of turgor pressure, rapid movement, and leaf folding, a behavior that was speculated to startle insects. Hagihara et al. has now shown that such responses are generated by rapid changes in calcium concentration and electrical signals that propagate from the leaflet to the pulvinar cell. Further, Hagihara et al. has provided evidence that a remarkable reduction in calcium influx by pharmacological modulators, as well as CRISPR-Cas9 genome editing technology, can block leaf movements and confer vulnerability to herbivorous insects. The experiments and conclusions are in favor of their proposed hypothesis. However, I have serious concerns regarding the pharmacological approaches used in this study.

Major points:

1) Ca^{2+} has been considered a long-distance defense signaling molecule. To determine if Ca^{2+} is required in plant-herbivore interactions, researchers often employ the Ca^{2+} signaling disruption approach. This strategy involves either restricting Ca^{2+} entrance or altering the quantity of accessible Ca^{2+} .

(A) Inhibition of Ca^{2+} entry can be achieved by using either 1) Non-Selective Ca^{2+} channel blockers such as Ca^{2+} channel blockers such as lanthanum (La^{3+}), gadolinium (Gd^{3+}) and Ruthenium Red or, 2) L-Type Calcium Channel Antagonists (verapamil, nifedipine, bepridil), $iGluR/GLR$ Antagonists (DNQX, CNQX, MNQX).

*In this study, Hagihara et al. have used La^{3+} to inhibit intracellular Ca^{2+} increase, electrical signaling, and rapid movement in *M. pudica*. While the results clearly supported the involvement of cytosolic Ca^{2+} in plant rapid movement, the use of La^{3+} is often associated with several side effects. La^{3+} has been known to induce phosphate deficiency; La^{3+} concentration greater than 100 μM caused changes in root architecture in the Arabidopsis plant (Ruiz-Herrera et al., Plant Soil 2012, 353, 231–247). When La^{3+} is exposed to low phosphate concentrations, it can create insoluble salts. Moreover, the activity of La^{3+} may be affected by the concentration of Ca^{2+} in the medium (Corzo et al., J. Gen. Microbiol. 1992,*

138, 1791–1795). It has also been shown that whereas La^{3+} inhibits calcium influx in the short term in response to low K^+ , 25 μM La^{3+} amplifies the long-term secondary Ca^{2+} signal caused by K^+ deprivation in *Arabidopsis* roots. (Behera et al., *New Phytol.* 2017, 213, 739–750). Hagihara et al. have used 50 mM La^{3+} without K^+ to inhibit leaf movement, electrical signaling in their experiments. Based on the previous observation, the authors need to exclude these La^{3+} side effects from their data and how they may affect the results. The authors could test the effect of lower concentrations of La^{3+} and not just 50 mM concentration to determine if there is a threshold level for La^{3+} inhibitory action.

We thank the reviewer for this suggestion. We performed additional experiments to address this critical question (new Extended Data Fig. 13 and 14). We prepared La^{3+} solution with 20 mM KNO_3 , 10 mM $\text{Ca}(\text{NO}_3)_2$, and 1 mM MES (pH adjusted to 5.8 with Tris) and used it to treat *M. pudica* leaves. This treatment inhibited the propagation of $[\text{Ca}^{2+}]_{\text{cyt}}$ and electrical signals and suppressed leaflet movements. We also conducted experiments to determine the threshold level for the inhibitory action of La^{3+} using the same solution ($\text{IC}_{50} = 0.52$ mM). To exclude the possibility that phosphate deficiency affects Ca^{2+} /electrical signaling, we performed similar experiments using verapamil, a voltage-dependent Ca^{2+} channel antagonist. Verapamil retarded the propagation of both Ca^{2+} and electrical signals in rachillae. We have added these data and revised the text as follows.

Revised main text: Page 6: “Our model is based on pharmacological methods with possible side effects; e.g., La^{3+} amplifies a secondary Ca^{2+} increase induced by 0 mM K^+ medium in *Arabidopsis thaliana* roots²², causes plasma membrane depolarization that is reduced by raising the extracellular Ca^{2+} concentration in *Neurospora crassa*²³, and induces phosphate deficiency by precipitating phosphate, which modulates the root system architecture in *A. thaliana*²⁴.”

“To address these critical questions, we conducted pharmacological experiments with different concentrations and chemicals. A La^{3+} solution with high K^+ and Ca^{2+} concentrations inhibited wound-induced $[\text{Ca}^{2+}]_{\text{cyt}}$ increases at the wound sites, leaflet veins, pulvini, and rachillae and subsequent leaflet movements (Extended Data Fig. 13a–c). The dose–response curve indicated that the half maximal inhibitory concentration (IC_{50}) of La^{3+} was 0.52 mM (Extended Data Fig. 13d). These data suggest that the inhibitory effects on $[\text{Ca}^{2+}]_{\text{cyt}}$ dynamics and leaflet movements were due to La^{3+} but not to the K^+ deficiency and membrane depolarization. To exclude the possibility that La^{3+} -induced phosphate deficiency hindered $[\text{Ca}^{2+}]_{\text{cyt}}$ and electrical signals, we used verapamil, another Ca^{2+} channel antagonist, which retarded Ca^{2+} /electrical signal propagation in the rachilla (Extended Data Fig. 14).”

“Therefore, the Ca^{2+} /electrical signal and leaflet movements are unlikely to be inhibited by the adverse effects of La^{3+} and EGTA.”

Revised Extended Data Fig. 13: “Extended Data Fig. 13 | La^{3+} solution with high K^+ and Ca^{2+} concentrations inhibited long-distance Ca^{2+} /electrical signal propagation and leaflet movements. **a, b** $[\text{Ca}^{2+}]_{\text{cyt}}$ signatures at tertiary pulvini and leaflet angle changes in leaves pretreated with control (**a**, $n = 6$) and 50 mM La^{3+} solution (**b**, $n = 6$). **c** $[\text{Ca}^{2+}]_{\text{cyt}}$ changes at the wound site (W), leaflet vein (V), pulvinus (P), and rachilla (R) ($n = 8$ each). For ROI positions, see Fig. 2b or Extended Data Fig. 3c. **d** Sigmoidal dose–response curve (black line). The probability of the electrical response at e1 (see Fig. 2h) relative to the control (red dots) was plotted against the La^{3+} concentration, for which IC_{50} was 0.52 mM.”

Revised Extended Data Fig. 14: “Extended Data Fig. 14 | Verapamil inhibits both $[\text{Ca}^{2+}]_{\text{cyt}}$ and electrical signals. **a, b** Wounding triggered $[\text{Ca}^{2+}]_{\text{cyt}}$ and electrical signals basipetally propagating in a rachilla in a control leaf (**a**) but not in a leaf treated with 2 mM verapamil (**b**). Typical data are displayed ($n = 8$ for **a**; $n = 5$ for **b**).”

Revised Methods text: Page 20: “ $\text{LaCl}_3 \cdot 7\text{H}_2\text{O}$ (Wako) was dissolved in ECF (Extended Data Fig. 5) in high K^+ and Ca^{2+} solution [20 mM KNO_3 , 10 mM $\text{Ca}(\text{NO}_3)_2$, 1 mM MES; pH adjusted to 5.8 with Tris(hydroxymethyl)aminomethane; Extended Data Fig. 13] or in water for other experiments at a final concentration of 0.1–50 mM.”

Revised Methods text: Pages 20–21: “Verapamil hydrochloride (Sigma) was dissolved in ECF at a final concentration of 2 mM. The petioles of the leaf specimens were dipped into and treated with these solutions in 15-mL tubes for 2–3 h for La^{3+} treatment, 3–5 h for EGTA treatment, or 5 h for verapamil treatment under white LED light. The leaves were then dipped in a control solution or 1 mM KCl for EGTA-treated leaves (Extended Data Fig. 6) to stop the pharmacological treatments and were allowed to recover for at least 1 h under white LED light”

Revised Methods text: Page 22: “ IC_{50} for the inhibitory effects of La^{3+} was estimated by fitting a sigmoidal dose-response curve.”

(B) *The alteration of the amount of available Ca^{2+} can be achieved by chelation with Ca^{2+} chelators, such as EGTA, EDTA, and BAPTA, which bind to Ca^{2+} ions with high affinity in a selective, reversible manner. In this study, the authors have used 50 mM EGTA to alter*

[Ca²⁺]_{cyt} increase and movement in response to wounding in M. pudica. While EGTA is a strong Ca²⁺ chelator, it does have certain obvious drawbacks that should be considered while designing an experiment. One feature of Ca²⁺ chelators is that they are not only in equilibrium with Ca²⁺ ions, but also with H⁺, as Ca²⁺ binding to a chelator typically results in the release of H⁺ (reducing the pH). The converse is likewise true for pH-sensitive chelators such as EGTA, where a change in pH significantly decreases the Ca²⁺ buffering capacity. Thus, it is critical to utilize a pH buffer in conjunction with the Ca²⁺ chelator in order to minimize the adverse effects of pH variations on the chelator's capacity to buffer Ca²⁺ ions. Additionally, prolonged EGTA treatment may alter the cell wall characteristics by decreasing the amount of Ca²⁺-bound pectin (Patton et al. Cell Calcium 2004, 35, 427–431). Notably, effective Ca²⁺ chelation with EGTA needs a pH of 8.0. Hagihara et al. have used 50 mM EGTA without any pH buffer for 5 hours. Therefore, it would be ideal to test the effect of EGTA prepared in a pH buffer, incubated for a shorter time period, and observe that the obtained result is still consistent.

This is an extremely important point. First, we tested 50 mM EGTA solution at pH 8.0 as suggested, but the control solution without EGTA (pH 8.0) inhibited Ca²⁺/electrical signal propagation in *M. pudica*. Therefore, we used 50 mM EGTA solution buffered with 10 mM HEPES (pH adjusted to 7.0 with KOH) and found that the EGTA solution buffered with HEPES retarded Ca²⁺/electrical signal transmission. We also reduced the period of EGTA treatment from 5 to 3 h. The Ca²⁺/electrical signal was inhibited in the leaves treated with EGTA and HEPES for the shorter period. We have added these data and rewritten the text as follows (new Extended Data Fig. 15).

Revised main text: Page 6: “In addition, prolonged EGTA treatment alters the mechanical strength of inflorescence stems in *Paeonia lactiflora* since EGTA removes Ca²⁺ from the cell wall²⁵.”

“We also reduced the period of EGTA treatment and mitigated the reduction in the Ca²⁺ buffering capacity of EGTA with the pH buffer HEPES²⁶. The [Ca²⁺]_{cyt} increase was limited within the wounded leaflets (Extended Data Fig. 15a, b), and the long-distance Ca²⁺/electrical signal was inhibited in the rachillae of leaves treated with EGTA and HEPES for a shorter period (Extended Data Fig. 15c–f).”

Revised Extended Data Fig. 15: “**Extended Data Fig. 15 | The long-distance [Ca²⁺]_{cyt}/electrical signal was inhibited by buffered EGTA solution. a, b** [Ca²⁺]_{cyt} changes at the wound sites (W), leaflet veins (V), pulvini (P), and rachillae (R) in leaves treated with 50 mM EGTA for 5 (a, n = 6) or 3 h (b, n = 6). For ROI positions, see Fig. 2b or Extended Data Fig. 3c. **c, d** Ca²⁺ and electrical signals were observed in the rachilla of

a leaf treated with control solution for 5 h (**c**) but not in a leaf treated with 50 mM EGTA solution for 5 h (**d**). The basipetal propagation of these signals was monitored. **e**, **f** Similar results were obtained when the treatment period was reduced from 5 to 3 h (**e**, control; **f**, EGTA). Representative data are displayed (**c** and **e**, $n = 5$; **d** and **f**, $n = 8$)."

Revised Methods text: Page 20: "EGTA (Dojindo Laboratories) was dissolved in water (Extended Data Fig. 3b–f, Extended Data Fig. 6) or 10 mM HEPES (Extended Data Fig. 15) to make a 50-mM solution, and the pH was adjusted to 7.0 with KOH and HCl."

"The petioles of the leaf specimens were dipped into and treated with these solutions in 15-mL tubes for 2–3 h for La^{3+} treatment, 3–5 h for EGTA treatment, or 5 h for verapamil treatment under white LED light. The leaves were then dipped in a control solution or 1 mM KCl for EGTA-treated leaves (Extended Data Fig. 6) to stop the pharmacological treatments and were allowed to recover for at least 1 h under white LED light."

3) For the quantification of Ca^{2+} imaging experiments, the authors have used 3-5 sample N values in many experiments. To bolster the finding, it would be preferable to conduct these experiments with a more number of sample N values to ensure that the results are more repeatable.

We very much thank the reviewer for this suggestion. We repeated several experiments as much as possible and revised the data and text as follows (Extended Data Fig. 2, 3, 5, 6 and 11, and Extended Data Table1). In addition, we acquired the data for Extended Data Fig. 2 again because the original data included only five samples and the graph was not clear. The conclusion was not changed by repeating the experiment. We replaced the data and table and rewrote the figure legend as indicated below.

Revised main text: Pages 3–4: "Wounding by scissors immediately elicited a $[\text{Ca}^{2+}]_{\text{cyt}}$ increase at the wound site, which subsequently propagated in a leaflet vein at 1.31 ± 0.17 mm/s ($n = 6$; Fig. 2a–c, Supplementary Movie 5)."

Revised Extended Data Fig. 2: "Leaflet angle changes were triggered by wounding leaflets and monitored over time in the WT (black, $n = 10$) and GCaMP6f leaves (red, $n = 10$). The third leaf from the top of the stem was cut from 2-month-old plants and used for this experiment."

Revised Extended Data Fig. 3: "**d–g** Wound-induced $[\text{Ca}^{2+}]_{\text{cyt}}$ changes at the wound site (**d**), leaflet vein (**e**), tertiary pulvinus (**f**), and rachilla (**g**) in leaves treated with 50 mM La^{3+}

and 50 mM EGTA ($n = 7$ each).”

Revised Extended Data Fig. 5: “Typical data are displayed ($n = 5$ for **a**, **b**, **e**, and **f**; $n = 6$ for **d** and **h**; and $n = 8$ for **c** and **g**).”

Revised Extended Data Fig. 6: “Typical data are displayed ($n = 4$ for **a** and **e**; $n = 5$ for **b** and **f**; and $n = 6$ for **c**, **d**, **g**, and **h**).”

Revised Extended Data Fig. 11: “Representative data are displayed ($n = 10$ for **a**; $n = 6$ for **b**; $n = 16$ for **c**; and $n = 12$ for **d**).”

Minor points:

- 1) *Inconsistency in the depiction of the Y-axis "time after wounding" timing: several graphs missing the 0-time point, for example, 1e, 2g, Extended Data Fig. 4b, Fig. 6e,f,g,h, Fig. 7e,f,g,h.*
- 2) *Gap missing in the Figure legend: Fig 1, $n=5$, $n=7$*
- 3) *Gap missing Line 77: $n=10$; Line 119: $n= 14$; line 132: $n= 12$*

We apologize for these oversights. We have edited graphs that lacked the 0-time points (Fig. 1e, Fig. 2j, Extended Data Fig. 4b, Extended Data Fig. 5e, f, g, h, Extended Data Fig. 6e, f, g, h, Extended Data Fig. 7, and Extended Data Fig. 12b, d) as suggested. We have also revised gap missing in the text as follows.

Revised main text: Pages 3–4: “Wounding by scissors immediately elicited a $[\text{Ca}^{2+}]_{\text{cyt}}$ increase at the wound site, which subsequently propagated in a leaflet vein at 1.31 ± 0.17 mm/s ($n = 6$; Fig. 2a–c, Supplementary Movie 5).”

“The $[\text{Ca}^{2+}]_{\text{cyt}}$ signature on the rachilla was bimodal (Fig. 2d), with the first $[\text{Ca}^{2+}]_{\text{cyt}}$ peak propagating basipetally at 3.11 ± 0.41 and acropetally at 2.65 ± 0.38 mm/s ($n = 10$ each).”

Revised main text: Pages 4–5: “Grasshopper herbivores stayed and fed on the La^{3+} -treated leaves more than on the control leaves (Fig. 3d). La^{3+} -treated leaves lost 38.0% in weight after this feeding assay, which was 2-fold higher consumption than that of the control leaves [19.0% ($n = 14$ each); Fig. 3f]. Consistent with this result, the total residence time of grasshoppers was 75.3 ± 11.3 min on the control leaves and 161.2 ± 23.1 min on the La^{3+} -treated leaves ($n = 14$ each; Fig. 3h).”

Revised main text: Page 5: “As seen in the immobile La^{3+} -treated leaves, the immobile *elp1b1elp1b2* leaves lost 30.3% in weight by grasshopper attacks, which was

approximately double that of wild-type (WT) leaves [15.3% ($n = 12$ each); Fig. 3e, g], and the total residence time of grasshoppers was 80.2 ± 12.4 min on the WT leaves and 142.6 ± 22.3 min on *elp1b1elp1b2* leaves ($n = 12$ each; Fig. 3i).”

Revised Fig. 1: “e, f $[Ca^{2+}]_{\text{cyt}}$ signatures at the tertiary pulvinus and leaflet angle in leaves pretreated with H_2O (e, $n = 5$) and 50 mM La^{3+} (f, $n = 7$). Scale bars, 5 mm (a and b) or 1 mm (c and d).”

Reviewer #2:

*The manuscript by Hagihara and co-workers deals with a fascinating topic, the underlying signalling processes leading to the extremely fast leaf(let) movements in Mimosa pudica upon touch and wounding, which is almost unique in plant kingdom. Further, they addressed the longstanding question of the ecological relevance of this movement. Using a combination of reporter gene-based (GCaMP6f) cytosolic Ca^{2+} imaging, electrophysiology, pharmacology, and CRISPR/Cas9 generated immotile mutants (*elp1b1elp1b2*) the authors provide a wide range of convincing results.*

*The study shows the involvement of both electrical signalling, i.e. action potential and variation potential, and transient cytosolic Ca^{2+} level increase in the wound induced leaf movements. Both signals propagate fast (few mm/sec) along the veins acro- and basipetally. Employing grasshoppers and herbivorous lepidopteran larvae it was demonstrated that immotile *M. pudica* plants are more susceptible to insect feeding compared with WT plants. The authors suggest that the presentation of trichomes after leaflet movements or leaves touching the insect body are responsible for the protection. Maybe the sudden strong movement of the leaf surface scares the insects in addition?*

The MS text, all figures and videos are presented very well; conclusions are justified by the results. Remaining open questions such as the nature of the molecular mechanisms driving the Ca^{2+} transients and/or the electrical signals and the channels that may be involved have already been formulated in the conclusions. Expecting answers to these questions in the present work would go far beyond the aim of the study.

To be honest, it was really fun to read this fantastic work.

We very much thank the reviewer for the encouraging comments. In this revision, we deleted the discussion regarding the possible roles of hairs (trichomes) in defense responses since the *elp1b* mutants are still hairy and, in any case many other factors could

determine the defenses of leaves as suggested by the other reviewer. However, we plan to create hair-less or slow-moving *M. pudica* using CRISPR/Cas9 and will investigate whether the presentation of trichomes after leaflet movements, leaves touching the insect body and “the sudden strong movement of the leaf surface” are responsible for the protection.

Reviewer #3:

The Editor, Nature communications March 24, 2022

*Re: Reviewing the manuscript: Calcium-mediated rapid movements defend against herbivorous insects in *Mimosa pudica*.*

Dear Editor,

I was very happy to review this manuscript, thanks for the opportunity. This paper is a class for itself. Very very well done. It was a great pleasure to carefully examine the videos and figures. With all my love to my original contribution to defensive plant movement, the paper you asked me to review is much better than anything published in this subject since Darwin's time! In general, there are only a few papers that showed actual defense from herbivory by plant movement.

I have no conflict of interests, and I don't know the authors.

(1) The most critical result of that paper is that the authors showed by blocking the leaflets movements that insects attacked the disabled leaves much more than the ones that can fold their leaflets. This is a wonderful demonstration of the defensive role of leaflet folding, arriving after many decades of theoretical discussions (including mine).

*(2) The use of CRISPER for disabling plant movement will show the way to many other scientists not only in arresting leaflet movement, but in many other types of functions. The use of chemical blockers to arrest leaflet movement is also impressive and very innovative concerning anti-herbivory defense. This paper opens new avenues. I hope that the authors will do the same with *Desmodium motorium* (the Indian telegraph plant) that I discussed theoretically.*

(3) From the above, it is clear that the paper is first class in originality. It is exciting to read it, and to think about many potential studies that can and will follow, and not only in defensive plant movements. They are the first, and I cannot compare it to other papers in defensive plant biology.

(4) The work supports the conclusions. The methods are excellent/brilliant and can be reproduced. There is nothing that is not OK, and I have only several small suggestions about the wording that I will list below.

(5) *As usual, my name Simcha Lev-Yadun should be revealed to the authors.*

I suggest changing several words.

(I) *I suggest to change line 113 to:leaves that did not respond to wounding or touch.....*

We very much thank the reviewer for this suggestion. We have corrected the main text as suggested.

Revised main text: Page 4: “To ask if these rapid movements do indeed serve as a defense response to wounding by insect herbivory, we used La³⁺-treated leaves that did not respond to wounding or touch (Fig. 3a, b, Extended Data Fig. 7, Extended Data Fig. 8a, b).”

(II) *I suggest to change line 115 to:more than on the control leaves.....treated leaves lost 38.0%...*

Again, we thank the reviewer for this suggestion. We have revised the main text as suggested.

Revised main text: Pages 4–5: “Grasshopper herbivores stayed and fed on the La³⁺-treated leaves more than on the control leaves (Fig. 3d). La³⁺-treated leaves lost 38.0% in weight after this feeding assay, which was 2-fold higher consumption than that of the control leaves [19.0% ($n = 14$ each); Fig. 3f].”

(III) *I suggest to change in line 133 the word "Thus" with "Since"*

We have revised the main text as follows.

Revised main text: Page 5: “This effect was not limited to grasshoppers since we also used a generalist caterpillar and obtained similar results (Extended Data Fig. 12b–e).”

(IV) *I suggest to change line 151 to:of the phytohormones ethylene and jasmonate.....*

We thank the reviewer for this suggestion. We have revised the main text accordingly and cited relevant references.

Revised main text: Page 5: “Plants activate local and systemic defense responses within minutes to hours of insect contact, wounding, or herbivory, e.g., through production of the phytohormones ethylene and jasmonate, priming non-damaged regions to mount pre-

emptive defenses¹⁸⁻²¹.”

Sincerely,

Dr. Simcha Lev-Yadun, Professor

Department of Biology & Environment, Faculty of Natural Sciences, University of Haifa-Oranim, Tivon 36006, Israel

Reviewer #4:

*The movements of *M. Pudica* continue to fascinate plant biologists. Much early literature has investigated the role of electrical signaling and Ca^{2+} in leaf movements in this plant. The current manuscript by Hagihara et al. applies (for the first time to my knowledge) the tools of molecular genetics to investigate the roles of stimulus-induced leaf movements in *M. pudica*. In these experiments, the authors generated *M. pudica* plants that lack functional pulvini and therefore lack the ability to move when touched or wounded. To do that they cleverly targeted and mutated LOB domain transcription factors known to be necessary for pulvinus formation in other members of the Fabaceae. This revealed that the leaves of these plants were eaten more quickly by insects than those of the wild type. This finding is perhaps the most innovative part of this manuscript.*

*In other experiments Hagihara et al. investigated the role of Ca^{2+} signaling in leaf movement. To do this *M. pudica* was transformed (again for the first time to my knowledge) with GCaMP6f. The overall findings are that pulvinar Ca^{2+} increases occur prior to leaflet movements. The new Ca^{2+} data go well beyond what has been published previous with non-transgenic approaches since they have high temporal resolution. However, data from parallel pharmacological experiment seem to be over-interpreted. This leads to statements that imply causality rather than correlation. Also, some important background and methods details are lacking. Overall, the manuscript is clearly written, but it gives the impression of being a mixture of different objectives (defense, calcium signaling, wounding, cold treatments and pharmacological experiments) that are not always linked well in the text. The videos are spectacular and are a useful addition-especially those showing simultaneous Ca^{2+} and electrical recordings.*

Major points

1. For each CRISPR-Cas9 line the authors show two mutated sequences (Extended data Figure 10). The authors cite reference 26 in the Methods section, but are they sure (given the common diversity of ploidy levels in *Mimosa* species) that the plants they used were tetraploid? The

authors should state clearly how many ELP1B genes there are in the plants they used and whether all copies of each gene were mutated.

We very much thank the reviewer for this suggestion. Sequencing results of *ELP1B1* and *ELP1B2* showed dual peak profiles, suggesting that each gene has two alleles. The data displayed in new Extended Data Fig. 9 and 10 are consistent with the idea that *M. pudica* is tetraploid²⁶. We have added the sequencing data for *ELP1B* genes (new Extended Data Fig. 9) and modified the Extended Data Fig. 10 to answer these questions.

Revised Extended Data Fig. 9: “Extended Data Fig. 9 | Sequencing chromatograms of *ELP1B1* and *ELP1B2* genes. Genomic DNAs were obtained from wild-type *M. pudica* (WT) or T₀ founders of the CRISPR/Cas9 transgenic lines of 8 independent origins (#1–6 and #8–9). Dashed lines indicate the position of Cas9 cleavage. Arrowheads indicate a discriminative nucleotide between *ELP1B1* and *ELP1B2* genes, which ensured specific PCR amplification of each single gene. Dual peak profiles detected in transgenic founders suggest the presence of two alleles for each gene. Signed numbers in the upper right of each chromatogram indicate the sizes of indels. It could not be distinguished whether the apparent single profile, for example in *ELP1B1* of #1, is ascribed to the same mutation in the two alleles or to a possible failure of PCR amplification caused by a large genomic deletion.”

2. Key experiments use EGTA or La3+ treatments to block Ca2+ signaling. These treatments block leaf movements and cytosolic Ca2+ increases. The authors conclude (lines 66, 67) that cytosolic calcium increases in the pulvinus cause rapid movements. However, correlation is not causation. La3+ (50 mM) or EGTA treatments are expected to block much of a cell's function. The authors can only conclude that blocking cytosolic calcium signaling correlates with inhibiting leaf movements.

We thank the reviewer for this suggestion. We have revised the text accordingly.

Revised main text: Page 3: “Therefore, a $[Ca^{2+}]_{cyt}$ increase in the pulvinus is correlated with rapid leaf movement.”

3. Lines 92-100. The cold treatments don't seem to fit in with the rest of the paper which concentrates on wounding and herbivory. If the cold treatment data is left in it should be better integrated in the text. For example, ED Figure 3 uses cold water treatments to separate APs and VPs. Do leaf movements still take place under the authors conditions? I think this is already known and the authors should check reference 4. If not, they should relate their data for

bimodal Ca²⁺ peaks and electrical signals seen after wounding (Figure 2) to the cold water experiments. Is only the AP necessary for leaf movement under their conditions? How are the cold treatments related to herbivory and defense?

We thank the reviewer for raising this important point. We have removed sentences and data associated with cold treatment and the refractory period because these data did not align with the manuscript. To gain deep insights into the distinct electrical signals AP and VP, we performed additional experiments on the touch response (new Fig 2) as presented in the Introduction section and Fig. 1a. We also discussed the potential different roles of touch and wound-induced Ca²⁺/electrical signaling in motion-based defense to better integrate the touch response data into the manuscript.

Revised main text: Page 4: “Since non-wounding stimuli trigger only an AP⁷ and mechanical wounding generates both an AP and VP propagating in a rachilla toward the pulvinus⁵, we investigated the spatiotemporal relationship between [Ca²⁺]_{cyt} transmission and the electrical signals. Touching a pinna tip evoked single-peak [Ca²⁺]_{cyt} signal and AP with leaflet movements in a rachilla (Fig. 2e, g, i, Extended Data Table 1). Wounding a leaflet triggered propagation of bimodal [Ca²⁺]_{cyt} and electrical signals consisting of an AP and VP (the first and second peaks, respectively) in a rachilla (Fig. 2f, h, j, Extended Data Fig. 4, Supplementary Movie 8). The touch-induced [Ca²⁺]_{cyt} signal and AP propagated on the rachilla at 5.87 ± 0.75 and 5.52 ± 0.43 mm/s, respectively (Extended Data Table 1). The wound-induced [Ca²⁺]_{cyt} signal and AP transmitted on the rachilla at 4.13 ± 0.45 and 4.27 ± 0.41 mm/s, respectively (Extended Data Table 1).”

“Moreover, in contrast to the control experiments (Extended Data Fig. 5a, b, e, f, Extended Data Fig. 6a, b, e, f, Supplementary Movies 9, 10, 13, 14), La³⁺ and EGTA pretreatments inhibited the bidirectional propagation of the [Ca²⁺]_{cyt} increases and APs/VPs (Extended Data Fig. 5c, d, g, h, Extended Data Fig. 6c, d, g, h, Supplementary Movies 11, 12, 15, 16), suggesting that the long-distance transmission of Ca²⁺ changes is spatiotemporally coupled with the AP and VP, triggering rapid leaf movements in touched or wounded *M. pudica*.”

Revised main text: Pages 5–6: “Touch triggered a single-peak [Ca²⁺]_{cyt} change and AP in the rachilla, sequentially inducing [Ca²⁺]_{cyt} increases at pulvini and subsequent leaflet movements (Fig. 1a, c, e, Fig. 2e, i). Thus, *M. pudica* might sense herbivore contacts to activate motion-based defenses before leaves are damaged, but leaflet movements are restricted within the touched pinna because the AP cannot propagate over the secondary pulvinus toward distant pinnae⁷. Wounding elicits a VP that can pass through primary and secondary pulvini^{5,7}, triggering defenses in both the local and systemic leaves.”

Revised Fig. 2: “e, f Simultaneous recording of $[Ca^{2+}]_{cyt}$ increases (yellow arrowhead) and electrical signals and leaflet movements (red arrowhead) caused by touch (e) or wounding (f) as indicated by white arrows (0 s). g, h Electrodes (e1 and e2, blue rectangles) and ROIs (red arrows, 1 mm from the electrodes) were set on the rachilla for surface potential measurement and $[Ca^{2+}]_{cyt}$ analysis, respectively. A pair of leaflets was numbered from the base of a pinna. The tip of a leaf pinna was touched by forceps (g), or leaflet number 12 was wounded with dissecting scissors (h, W). i, j Changes in $[Ca^{2+}]_{cyt}$ and surface potential in response to touch (i) or wounding (j) (colors as depicted in g or h).”

Other points

Line 54 ‘To determine how’ ... Here, since the authors do not generate mechanistic insights, it would be more realistic to write ‘To determine whether...’

We thank the reviewer for this suggestion. We have revised the main text accordingly.

Revised main text: Page 3: “To determine whether rapid movements might be regulated by Ca^{2+} and electrical signals, we created transgenic *M. pudica* expressing the genetically encoded Ca^{2+} indicator GCaMP6f¹⁶ and visualized the spatiotemporal dynamics of the cytosolic Ca^{2+} concentration ($[Ca^{2+}]_{cyt}$) in real time.”

Lines 96, 97. The text, which relates to Extended Data Figure 5c and 5d, is not clear to me. Figure 5c implies that consecutive cold treatments on the same sample plant were used to estimate refractory periods. In Figure 5c, after an initial cold-induced AP the next event is produced after 2, plus 3 min, plus 4 min, plus 5.3 min (= 14.3 min). It is usual to establish refractory periods with a single time interval between the first and last cold treatment.

As stated in the response to point 3, we have removed sentences and data related to cold treatment and the refractory period.

*Lines 110, 111 The authors cite three references for the putative antiherbivore defense role of leaf movements in *M. pudica*. Please note that this idea greatly pre-dates the references cited. The authors should cite reference 2 (Bose, 1926) here.*

We apologize for this oversight. We have corrected the text accordingly.

Revised main text: Page 3: “Although numerous studies have assumed physiological

roles of these rapid movements, e.g., being unnoticed against the dark ground², startling insects¹¹, exposing thorns¹², and giving the appearance of a less voluminous meal¹³, clear evidence supporting these theories thus far does not exist.”

Fig. 2c The number of replicates should be given. There is no visible error envelope for the leaflet vein data. Is this because few replicates were used?

We apologize for the oversight. As also explained to Reviewer #1, the number of replicates was small. Therefore, we repeated the experiments and updated the data (Fig. 2, Extended Data Fig. 2, 3, 5, 6 and 11, and Extended Data Table1). We have included the number of replicates in the figure legend. We have also added an explanation for why the $\Delta F/F_0$ curves were terminated at the time points before the end of the graph.

Revised Fig. 2: “**c, d** $[Ca^{2+}]_{\text{cyt}}$ changes monitored in the W, V, P (**c**, $n = 6$), and R regions (**d**, $n = 10$). The $\Delta F/F_0$ curves were terminated at the time points at which ROIs on W or V could not be traced because of leaflet movements.”

Fig. 2d The number of replicates should be given.

We apologize for this oversight. We have added the number of replicates in the figure legend as previously indicated.

Extended data Fig. 10 The authors should give the exact positions of the mutations relative to the predicted transcription start sites.

We apologize for this oversight. We have added the exact positions of the mutations relative to the translation start sites (Extended Data Fig. 10) because it is difficult to determine the transcription start sites precisely. We have also added the deduced amino acid sequences of the remnant ELP1B proteins in Extended Data Fig. 10.

Revised Extended Data Fig. 10: “**b** Pairwise alignment of ELP1B1 and ELP1B2 proteins. Amino acid residues conserved between the sequences are highlighted with a black background. **c** Deduced amino acid sequences of remnant ELP1B proteins in *elp1b1elp1b2* double mutants. In T₀ founder #1, all detected mutations were expected to cause frameshift and truncation of the gene products. In T₀ founder #8, 3 of 4 alleles had a 3-bp deletion, which does not cause frameshift and only affects 1 or 2 codons around the mutation.”

Extended data Fig. 13 This figure does not seem to fit in with any other results or observations

*and could be removed. Hairs on petioles seem irrelevant here since the authors look at damage to leaflet on rachillae. The *elp1b* mutants are still likely to be hairy and, in any case many other factors could determine the defenses of leaves.*

We thank the reviewer for this suggestion. We have removed the sentences, data, and a reference associated with hairs as suggested.

In the Methods section the text needs improvement:

Line 337 'was a bit complicated' should be removed

Line 391 'threes'

Line 454 'were recovered in ECF'. What does 'recovered' mean here?

We apologize for these oversights. We have revised these sentences accordingly.

Revised Methods text: Pages 16–17: “The entire construction procedure of this vector reflects a series of modifications for functional improvement.”

Revised Methods text: Page 18: “These three fragments were incorporated into the *AgeI*–*NheI* site of pSB11CAS11-ELP1A to replace the guide RNA cassette.”

Revised Methods text: Page 20: “The leaf specimens were treated as described in the Ca^{2+} imaging section using ECF instead of water.”

Reviewer #1 (Remarks to the Author):

I'd like to thank the Authors for their time in responding to my queries and considering my feedback. All of my concerns and suggestions (as well as the other reviewers) have been satisfactorily addressed by the authors. Additional material and a full description of the experiments have been added to the text by the authors, which has improved the clarity of the paper. This is an important addition to the "in vivo" calcium signaling research in plants and it will be beneficial to plant physiology and plant-herbivore interaction research. However, I do have one concern that needs clarification.

This paper shows the involvement of cytosolic Ca²⁺ in *M. pudica* plant rapid movement in response to herbivory. By using EGTA to chelate Ca²⁺ ions, the authors show that the suppression of cytosolic Ca²⁺ affects the process. However, EGTA is not membrane permeable, and the charges in EGTA prevent the chelator from entering the cytoplasm and chelating cytosolic Ca²⁺. Thus, the reduction of Ca²⁺ propagation in *M. pudica* leaves by EGTA implies that extracellular Ca²⁺ is involved in rapid movement and the plant's defensive responses. However, prior research has demonstrated that the interaction of plasma membrane and vacuolar ion channels causes rapid increases in cytosolic calcium in response to herbivory (Vincent et al. *Plant Cell*. 2017 6:1460-1479). Therefore the author needs to justify whether EGTA effectively chelates both cytosolic and apoplasmic Ca²⁺. This is especially important since the author used 5 hours with 50 mM EGTA, and this long incubation time might induce off-target effects.

To address this issue, the authors could use acetoxymethyl ester forms of the chelators EGTA-AM to completely chelate both cytosolic and apoplasmic Ca²⁺. Using this membrane-permeable EGTA-AM version would reduce the incubation time and improve the reliability of this experimental design.

Apart from this concern, the author's work provides a great deal of interesting information and insight into plant defense development.

Reviewer #4 (Remarks to the Author):

My comments on the first submission, including those concerning replication and the number of ELPB1 genes in *Mimosa pudica* have been addressed. The authors have produced an original and innovative paper on leaf movements in this plant. Hagihara et al. show that cytosolic Ca²⁺ levels increase in and near pulvini in response to touch, wounding and insect damage and that this is coupled to electrical signaling. By generating plants lacking pulvini, Hagihara et al. demonstrate the role of leaf movements in plant defense. The videos that accompany the manuscript are spectacular.

Minor point that should be corrected

Concerning lines 183-184: formally, GLRs are not Ca²⁺ channels.

Instead of '...ligand-gated Ca²⁺ channels..' it would be more correct to write '...ligand-gated ion channels..'

Reply to Reviewers

We are grateful to the reviewers for their encouraging comments and thoughtful reviews that helped us significantly improve our manuscript. We have addressed all the points raised, and specific responses for each query are detailed below:

Reviewer #1 (Remarks to the Author):

I'd like to thank the Authors for their time in responding to my queries and considering my feedback. All of my concerns and suggestions (as well as the other reviewers) have been satisfactorily addressed by the authors. Additional material and a full description of the experiments have been added to the text by the authors, which has improved the clarity of the paper. This is an important addition to the "in vivo" calcium signaling research in plants and it will be beneficial to plant physiology and plant-herbivore interaction research. However, I do have one concern that needs clarification.

*This paper shows the involvement of cytosolic Ca^{2+} in *M. pudica* plant rapid movement in response to herbivory. By using EGTA to chelate Ca^{2+} ions, the authors show that the suppression of cytosolic Ca^{2+} affects the process. However, EGTA is not membrane permeable, and the charges in EGTA prevent the chelator from entering the cytoplasm and chelating cytosolic Ca^{2+} . Thus, the reduction of Ca^{2+} propagation in *M. pudica* leaves by EGTA implies that extracellular Ca^{2+} is involved in rapid movement and the plant's defensive responses. However, prior research has demonstrated that the interaction of plasma membrane and vacuolar ion channels causes rapid increases in cytosolic calcium in response to herbivory (Vincent et al. *Plant Cell*. 2017 6:1460-1479). Therefore the author needs to justify whether EGTA effectively chelates both cytosolic and apoplastic Ca^{2+} . This is especially important since the author used 5 hours with 50 mM EGTA, and this long incubation time might induce off-target effects.*

To address this issue, the authors could use acetoxymethyl ester forms of the chelators EGTA-AM to completely chelate both cytosolic and apoplastic Ca^{2+} . Using this membrane-permeable EGTA-AM version would reduce the incubation time and improve the reliability of this experimental design.

Apart from this concern, the author's work provides a great deal of interesting information and insight into plant defense development.

We thank the reviewer for this suggestion. We performed additional experiments to address this critical question. Given that sufficient EGTA-AM was unavailable, we used BAPTA-

AM, together with EGTA, to chelate both cytosolic and apoplastic Ca^{2+} . To reduce the incubation time, we treated these chelators to the rachillae and shortened the incubation time to 0.5 h. In this new condition, the long-distance transmission of Ca^{2+} and electrical signals in rachillae was not triggered by wounding; however, these signals were found in control. These data support our model that the $[\text{Ca}^{2+}]_{\text{cyt}}$ changes coupled with electrical signals act as the long-distance signal triggering leaf movements. We have added these data in new Extended Data Fig. 16 and revised the texts as follows.

Revised main text: Page 6: “Furthermore, we treated membrane-permeable cytosolic Ca^{2+} chelator (BAPTA tetraacetoxymethyl ester) together with EGTA to the rachillae. Despite the shortening of the incubation period to 0.5 h, wounding did not induce the long-distance propagation of the Ca^{2+} and electrical signals in the rachillae (Extended Data Fig. 16). Therefore, the Ca^{2+} /electrical signal and leaflet movements are unlikely to be inhibited by the adverse effects of La^{3+} and EGTA, supporting our model that the $[\text{Ca}^{2+}]_{\text{cyt}}$ changes coupled with the electrical signals act as the long-distance signal triggering leaf movements.”

Revised Methods text: Page 21: “BAPTA tetraacetoxymethyl ester (BAPTA-AM, Tokyo Chemical Industry) was dissolved in DMSO to make a 100-mM stock solution. The stock solution was diluted 100 times using 10 mM HEPES solution with 50 mM EGTA, and the pH was adjusted to 7.0 using KOH and HCl. Pinnae were isolated from the leaves, and basal 7 leaflet pairs were removed from the base of the isolated pinnae to insert the pinnae into the solution. The isolated pinnae were inserted into 1.5-mL tubes filled with water through holes on the lids, fixed with Parafilm to the tubes, and placed in an incubator (LU-113, ESPEC) at 25°C overnight. The pinnae were transferred into PCR tubes with 0.3 mL of EGTA and BAPTA-AM, fixed to the tubes with Parafilm, and treated with the chelators under white LED light in the incubator at 25°C for 0.5 h. The treated pinnae were then put via holes in the lids into 1.5-mL tubes filled with 10 mM HEPES (pH was adjusted to 7.0 with KOH and HCl), fixed to the tubes with Parafilm, and maintained at approximately 25°C for more than 1 h under white LED illumination. Subsequently, the Ca^{2+} and electrical signals were measured (for positions of ROIs and recording electrodes, see Extended Data Fig. 16a).”

Revised Extended Data Fig. 16: “**Extended Data Fig. 16 | Co-treatment of EGTA and BAPTA-AM to rachillae.** **a** Electrodes (e1 and e2, blue rectangles) and ROIs (red arrows, 1 mm from the electrodes) were set on a rachilla for surface potential measurements and $[\text{Ca}^{2+}]_{\text{cyt}}$ analysis, respectively. A pair of leaflets was numbered from the base of a pinna. W, wounding. **b, c** Wounding triggers Ca^{2+} and electrical signals in the rachilla treated with

control solution for 0.5 h (**b**), but not in the rachilla treated with 50 mM EGTA and ~1 mM BAPTA-AM for 0.5 h (**c**). Representative data are displayed (**b**, $n = 5$; **c**, $n = 6$).”

Reviewer #4 (Remarks to the Author):

My comments on the first submission, including those concerning replication and the number of ELPB1 genes in Mimosa pudica have been addressed. The authors have produced an original and innovative paper on leaf movements in this plant. Hagihara et al. show that cytosolic Ca²⁺ levels increase in and near pulvini in response to touch, wounding and insect damage and that this is coupled to electrical signaling. By generating plants lacking pulvini, Hagihara et al. demonstrate the role of leaf movements in plant defense. The videos that accompany the manuscript are spectacular.

Minor point that should be corrected

Concerning lines 183-184: formally, GLRs are not Ca²⁺ channels.

Instead of '...ligand-gated Ca²⁺ channels..' it would be more correct to write '...ligand-gated ion channels..'

We apologize for this oversight. We have revised the main text as suggested.

Revised main text: Page 7: “Since VPs are related to the chemicals released upon wounding and transported through the xylem^{1,30}, ligand-gated ion channels, e.g., *GLUTAMATE RECEPTOR-LIKE* family, might be involved in *M. pudica*’s VP³¹.”

Reviewer #1 (Remarks to the Author):

All of my concerns are addressed satisfactorily by the authors. This study will have a significant impact on the field, and provides an insightful perspective on how *mimosa pudica* uses calcium in addition to electrical signalling to mediate responses to stimuli.